# HiCache: A Plug-in Scaled-Hermite Upgrade for Taylor-Style Cache-then-Forecast Diffusion Acceleration

Liang Feng[1,2,†], Shikang Zheng[1,3,†], Jiacheng Liu[1], Yuqi Lin[1], Qinming Zhou[1,4], Peiliang Cai[1],
Xinyu Wang[1], Junjie Chen[1], Chang Zou[1], Yue Ma[5], Linfeng Zhang[1]

[1] Shanghai Jiao Tong University     [2] Fudan University     [3] South China University of Technology

[4] Tsinghua University     [5] Hong Kong University of Science and Technology

## ABSTRACT

Diffusion models have achieved remarkable success in content generation but suffer from prohibitive computational costs due to iterative sampling. While recent feature caching methods tend to accelerate inference through temporal extrapolation, these methods still suffer from severe quality loss due to the failure in modeling the complex dynamics of feature evolution. To solve this problem, this paper presents HiCache (**H**erm**i**te Polynomial-based Feature **Cache**), a training-free acceleration framework that fundamentally improves feature prediction by aligning mathematical tools with empirical properties. Our key insight is that feature derivative approximations in Diffusion Transformers exhibit multivariate Gaussian characteristics, motivating the use of Hermite polynomials—the potentially theoretically optimal basis for Gaussian-correlated processes. Besides, we introduce a dual-scaling mechanism that ensures numerical stability while preserving predictive accuracy, which is also effective when applied standalone to TaylorSeer. Extensive experiments demonstrate HiCache's superiority: achieving $5.55\times$ speedup on FLUX.1-dev while exceeding baseline quality, maintaining strong performance across text-to-image, video generation, and super-resolution tasks. Moreover, HiCache can be naturally added to the previous caching methods to enhance their performance, *e.g.,* improving ClusCa from 0.9480 to 0.9840 in terms of image rewards. *Code: https://github.com/fenglang918/HiCache.*

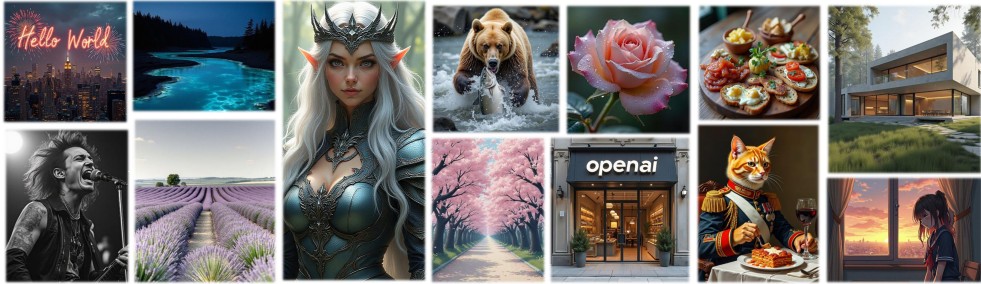

Figure 1: Teaser of HiCache using the FLUX.1-dev model, with a **6.24× FLOPs speedup** .

## 1 INTRODUCTION

Diffusion Models (DMs)  (Ho et al., 2020; Nichol & Dhariwal, 2021; Dhariwal & Nichol, 2021), especially when implemented with the scalable Transformer architecture (DiT)  (Peebles & Xie, 2023), have set a new standard for high-fidelity content generation  (Chen et al., 2023; Rombach et al., 2022; Karras et al., 2022; Saharia et al., 2022; Balaji et al., 2022). This success, however,

† Equal contribution. Emails: windbright918@gmail.com; zhanglinfeng@sjtu.edu.cn.
This project is sponsored by CCF-Baidu Open Fund.

is tempered by the substantial computational overhead inherent to the diffusion process. The iterative sampling, requiring hundreds of sequential forward passes through large models, results in high latency that critically hinders practical deployment, especially on diffusion transformers. Consequently, developing acceleration methods for diffusion models has become a hot research topic.

Among various acceleration strategies, feature caching has emerged as a particularly effective and training-free approach. Motivated by the observation that diffusion models exhibit similar features in the adjacent timesteps, the original feature caching seeks to directly reuse features in the adjacent timesteps (e.g., Deep-Cache (Ma et al., 2023), FORA (Selvaraju et al., 2024), ToCa (Zou et al., 2025), ClusCa (Zheng et al., 2025)), which leads to significant generation quality loss when the acceleration ratios increase. Recently, the paradigm has advanced to a "cache-then-forecast" mechanism as Tay-

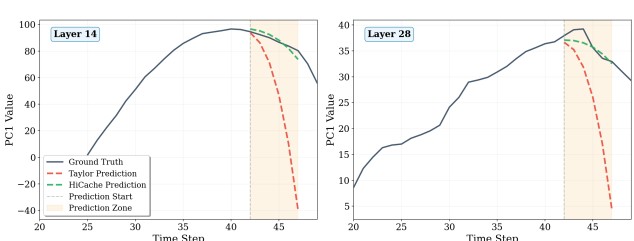

Figure 2: **Trajectory prediction comparison of Taylor and HiCache methods on FLUX.** *"Full trajectory"* indicates the feature trajectory from the original Flux (Esser et al., 2024) in the 14th and 28th layer. The y-axis denotes the most principal component of the features in diffusion models.

lorSeer (Liu et al., 2025a), where Taylor series expansion is used to extrapolate features in the future timesteps, significantly reducing the error from feature caching. TaylorSeer significantly reduces the error introduced by feature reusing, but still faces fundamental theoretical limitations: *The standard polynomial basis of a Taylor series is suboptimal for modeling the complex, non-monotonic trajectories of feature evolution in DMs.*

Figure 2 visualize the features of the original diffusion models (*i.e., ground-truth*) and the predictive features from TaylorSeer. It is observed that Taylor-based extrapolation exhibits significant deviations from ground truth trajectories, particularly at turning points where its monotonic nature fails to capture the underlying dynamics. These deviations indicate a modeling mismatch: the monomial Taylor basis poorly fits the non-monotonic, turning-point behavior of feature trajectories, causing approximation error to grow rapidly with prediction horizon and order (Proposition 1); this, in turn, caps the attainable acceleration before quality degradation becomes evident.

To overcome this limitation, in this paper, we propose HiCache (**H**ermite Polynomial-based Feature **Cache**), a framework that improves the feature forecasting by aligning the predictive basis with the intrinsic statistical properties of the feature dynamics. Our approach is based on a key empirical observation: *the derivative approximations of feature vectors in DiTs consistently exhibit a multivariate Gaussian distribution (details in Section 3.3.1)*. Motivated by this finding, HiCache replaces the standard polynomial basis with **scaled Hermite polynomials**. According to approximation theory (Szegö, 1975), Hermite polynomials are the potentially optimal orthogonal basis for representing Gaussian-correlated processes, making them theoretically better suited for this task. Besides, to address the numerical challenges of Hermite polynomials at large extrapolation steps, we further introduce a dual-scaling mechanism that simultaneously constrains predictions within the stable oscillatory regime and suppresses exponential coefficient growth in high-order terms through a single hyperparameter. This principled design enables both accurate and numerically stable feature prediction. As shown in Figure 2, the feature trajectory predicted by HiCache closely matches the original FLUX, demonstrating its effectiveness. Importantly, HiCache naturally serves as a drop-in replacement for Taylor-based predictors in existing cache-then-forecast frameworks, requiring only the substitution of the polynomial basis while preserving the same predictor form and computational structure with negligible overhead.

Extensive experiments demonstrate the effectiveness of HiCache in class-to-image generation with DiT, text-to-image generation with FLUX.1-dev, image-to-image generation with Inf-DiT for image super-resolution, and text-to-video generation with HunyuanVideo (Kong et al., 2024), showing improvements over previous methods. For instance, on FLUX.1-dev, HiCache achieves $5.55\times$ acceleration with $4.25\%$ improvement on ImageReward over the second-best method. This superiority extends to its use as a component upgrade: replacing ClusCa's Taylor predictor with our Hermite basis yields ImageReward improvements from $0.9480$ to $0.9840$ at $\mathcal{N} = 7$.

In summary, our main contributions are as follows.

- **Hermite Polynomial-based Feature Caching.** We propose a training–free feature extrapolation/cache predictor grounded in the empirically Gaussian finite–difference statistics of diffusion features; instantiated via KL–optimal scaled Hermite polynomials under Gaussian correlation, it better captures non–monotonic trajectories while preserving the Taylor–style form and compute.

- **Dual-scaling mechanism.** We introduce a single-hyperparameter dual scaling (input contraction + coefficient suppression) that stabilizes predictions and suppresses high-order growth without accuracy loss; it also works standalone for TaylorSeer.

- **Plug-and-play generality.** HiCache is model-agnostic and training-free; by only swapping the polynomial basis, it drops into any Taylor-style cache-then-forecast pipeline (e.g., TaylorSeer, ClusCa) with unchanged predictor form and negligible compute overhead—only a few per-step scalar basis-function evaluations and no extra matrix multiplications.

- **Broad empirical gains.** Across text-to-image, text-to-video, class-conditional, and super-resolution, we achieve equal-or-higher speedups with better quality: e.g., on FLUX.1-dev, $5.55\times$ speedup with ImageReward improving from $0.9872$ to $0.9979$; a zero-extra FLOPs basis swap lifts ClusCa from $0.9480$ to $0.9840$.

## 2 RELATED WORK

**Sampling Step Reduction** aims to reduce denoising iterations. DDIM (Song et al., 2021) introduced deterministic sampling with large steps, while DPM-Solver (Lu et al., 2022) formalized as an ODE with high-order numerical methods. Advances include knowledge distillation (Meng et al., 2023) and Consistency Models (Song et al., 2023) for few-step generation. Other approaches include cascaded diffusion models (Ho et al., 2022a) and improved sampling techniques (Karras et al., 2022). Text-to-image diffusion models show progress (Saharia et al., 2022; Balaji et al., 2022; Ramesh et al., 2022; Podell et al., 2023; Zhang et al., 2023), while video generation (Singer et al., 2022; Ho et al., 2022b) presents additional challenges.

**Denoising Network Compression** focuses on reducing per-step computational cost. Model compression methods include neural network pruning (Fang et al., 2023) and quantization (Li et al., 2023) have applied the traditional model compression methods to diffusion models. Besides, token merging (Bolya & Hoffman, 2023) has been introduced to reduce the number of tokens computed in each timestep. However, these methods tend to lead clear loss in generation quality, and sometimes rely on retraining the diffusion model.

**Feature Caching** has been recently proposed to offer a training-free alternative. Motivated by the great similarity of features in the adjacent timesteps, these methods aim to skip the computation in some timesteps based on the historical features, which can be roughly divided into two paradigms: (1) *Cache-then-Reuse* directly reuses features from adjacent timesteps (DeepCache (Ma et al., 2023), FORA (Selvaraju et al., 2024), ToCa (Zou et al., 2025), L2C (Ma et al., 2024)), but suffers quality degradation at large intervals; (2) *Cache-then-Forecast* predicts future features using historical data. TaylorSeer (Liu et al., 2025a) pioneered this approach with Taylor series expansion, significantly improving quality at high acceleration ratios. However, its standard polynomial basis remains suboptimal for complex feature dynamics, motivating our Hermite polynomial-based approach.

## 3 METHOD

### 3.1 PRELIMINARIES

Diffusion Transformers (DiT) (Peebles & Xie, 2023) apply transformer blocks iteratively during denoising, with each block updating features via $\mathcal{F}_{\text{out}} = \mathcal{F}_{\text{in}} + \text{MHSA}(\text{AdaLN}(\mathcal{F}_{\text{in}}, t)) + \text{MLP}(\text{AdaLN}(\cdot, t))$. The timestep-conditioned normalization and nonlinearities create complex, non-monotonic feature trajectories that challenge existing acceleration methods.

## 3.2 TAYLOR EXPANSION-BASED FEATURE PREDICTION

To accelerate sampling, existing methods like TaylorSeer extrapolate future features using Taylor series. We first formalize the finite difference framework:

**Definition 1** (Finite Difference Operator). *The $i$-th order backward difference operator $\Delta^{(i)}$ on feature trajectory $\{\mathcal{F}_t\}$ is defined recursively:*

$$\Delta^i \mathcal{F}_t = \frac{\Delta^{i-1} \mathcal{F}_t - \Delta^{i-1} \mathcal{F}_{t-N_{interval}}}{N_{interval}}, \quad with \quad \Delta^0 \mathcal{F}_t = \mathcal{F}_t \tag{1}$$

Taylor expansion-based prediction then takes the form:

$$\hat{\mathcal{F}}_{t-k}^{\text{Taylor}} = \mathcal{F}_t + \sum_{i=1}^{m} \frac{\Delta^i \mathcal{F}_t}{i!} (-k)^i \tag{2}$$

**Proposition 1** (Limitation of Monomial Basis). *For feature trajectories with bounded variation but containing turning points, the Taylor prediction error grows as:*

$$\|\hat{\mathcal{F}}_{t-k}^{Taylor} - \mathcal{F}_{t-k}\| = O\left( \frac{k^{m+1}}{(m+1)!} \sup_{\xi \in [t-k,t]} \|\mathcal{F}^{(m+1)}(\xi)\| \right) \tag{3}$$

*where the supremum can be arbitrarily large at trajectory inflection points.*

This limitation motivates our search for a basis that naturally captures non-monotonic behaviors.

## 3.3 HICACHE: HERMITE-BASED FEATURE CACHING

Our key insight is that feature differences in diffusion transformers exhibit Gaussian properties (empirically validated in Section 5), which theoretically motivates the use of Hermite polynomials as optimal basis functions.

**Proposition 2** (Gaussianity of Feature Differences). *Feature differences $\Delta F_t = F(x_t, t) - F(x_{t-\Delta}, t - \Delta)$ in diffusion transformers exhibit approximate Gaussian behavior through: (i) local linearization yielding conditional Gaussianity $\Delta F_t \approx \mathcal{N}(\mu_t, \Sigma_t)$ for small $\Delta$, and (ii) aggregation effects across network components invoking CLT with Berry-Esseen bound $O(\eta_t)$. See Appendix 4 for the complete statement and proof.*

**Corollary 1** (Optimal Basis Selection). *When the temporal correlation can be approximated by a Gaussian kernel $K(s, t) = \exp(-(s-t)^2/2\tau^2)$, the Karhunen-Loève expansion yields (scaled) Hermite functions as eigenfunctions. Thus, under such Gaussian kernel correlation, Hermite polynomials form the optimal orthogonal basis in the weighted $L^2(\gamma)$ sense. For general Gaussian processes, they remain approximately optimal (see Appendix 4, Theorem 2, for precise conditions).*

### 3.3.1 HERMITE POLYNOMIALS AS BASIS FUNCTIONS

Given the Gaussian nature, we adopt Hermite polynomials $H_n(x) = (-1)^n e^{x^2} \frac{d^n}{dx^n} e^{-x^2}$ (Abramowitz & Stegun, 1964) satisfying the recurrence $H_{n+1}(x) = 2x H_n(x) - 2n H_{n-1}(x)$. Unlike monotonically growing Taylor basis functions, Hermite polynomials exhibit oscillatory behavior (as shown in Figure 3b). This oscillatory nature provides implicit regularization (van der Westhuizen & Lasenby, 2018), enabling accurate capture of non-monotonic dynamics in neural network optimization.

Unlike monotonic Taylor basis, Hermite polynomials' oscillatory nature captures turning points. However, they face numerical instability for large arguments, motivating our scaled version:

**Definition 2** (Scaled Hermite Basis). *Scaled Hermite polynomials with contraction factor $\sigma \in (0, 1)$ are defined as:*

$$\tilde{H}_n(x) = \sigma^n H_n(\sigma x) \tag{4}$$

*This scaling provides dual stabilization: (i) input scaling $\sigma x$ constrains predictions within the stable oscillatory regime, and (ii) coefficient scaling $\sigma^n$ suppresses exponential growth in high-order terms.*

*Remark.* The dual-scaling mechanism can also be applied as a standalone enhancement to TaylorSeer, providing measurable gains without changing its basis.

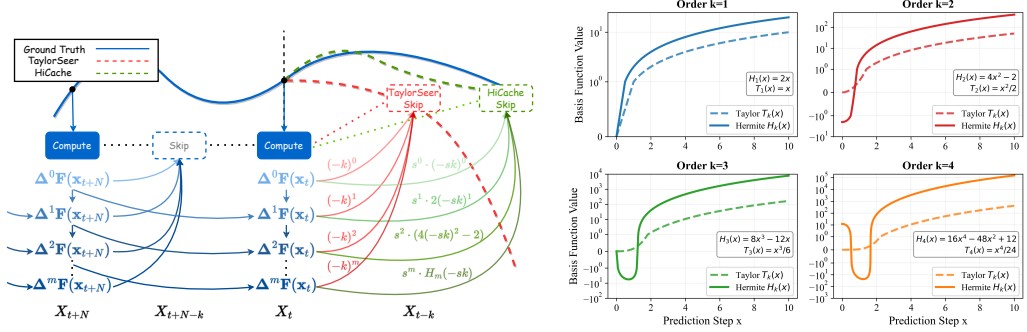

(a) Overview of HiCache compared with TaylorSeer.

(b) Comparison of basis functions.

Figure 3: **Method overview and basis function comparison.** (a) TaylorSeer (orange) predicts features using a power basis, whereas HiCache (green) keeps the functional form but swaps the basis for scaled Hermite polynomials. (b) Hermite's oscillatory behavior (e.g., $H_2(x)$ negative offset) captures non-monotonic evolution better than monotonic Taylor growth.

---

**Algorithm 1** HiCache: Hermite-based Feature Caching

---

**Require:** Interval $N_{\text{interval}}$, order $N_{\text{order}}$, contraction $\sigma$
1: Initialize $t_{\text{last}} \leftarrow T$ {Track last activation step}
2: **for** timestep $t = T$ to $1$ **do**
3:     **if** $t \mod N_{\text{interval}} = 0$ **then**
4:         Compute $\mathcal{F}_t = \Phi_t(x_t)$ {Full model forward}
5:         Update cache: $\{\Delta^i \mathcal{F}_t\}_{i=0}^{N_{\text{order}}}$ via Def. 1 with $\Delta t_{\text{hist}} = N_{\text{interval}}$
6:         $t_{\text{last}} \leftarrow t$ {Update last activation step}
7:     **else**
8:         $k \leftarrow t_{\text{last}} - t$ {Distance from last activation}
9:         Predict $\hat{\mathcal{F}}_t$ using Eq. (5) with $k$
10:     **end if**
11: **end for**

---

### 3.3.2 HICACHE ALGORITHM

**Proposition 3** (HiCache Feature Prediction). *Given cached derivative approximations $\{\Delta^i \mathcal{F}_t\}_{i=0}^{N_{order}}$, under Gaussian feature dynamics, a theoretically-motivated predictor is given by:*

$$\hat{\mathcal{F}}_{t-k}^{HiCache} = \mathcal{F}_t + \sum_{i=1}^{N_{order}} \frac{\Delta^i \mathcal{F}_t}{i!} \tilde{H}_i(-k) \tag{5}$$

*where $k \in \{1, \ldots, N_{interval} - 1\}$ is the prediction horizon and $\tilde{H}_i$ is the scaled Hermite polynomial from Definition 2. When coefficients are obtained via weighted least squares, this achieves projection-optimality in $L^2(\gamma)$. The direct use of $\Delta^i \mathcal{F}_t/i!$ provides an efficient approximation.*

HiCache reduces cost by factor $(N_{\text{interval}} - 1)/N_{\text{interval}}$ with $O(N_{\text{order}})$ overhead. The total error $\mathbf{E}_{\text{total}} = \mathbf{E}_{\text{truncation}} + \mathbf{E}_{\text{approximation}} + \mathbf{E}_{\text{numerical}}$ satisfies:

$$\|\mathbf{E}_{\text{total}}\| \leq O\left(\frac{(\sigma\sqrt{2}|\Delta s|)^{N+1}}{\sqrt{(N+1)!}}\right) + O(\Delta t_{\text{hist}}\sqrt{N}) + O(\epsilon_{\text{machine}}) \tag{6}$$

Under $\sigma\sqrt{2}|\Delta s| < 1$, the Hermite truncation $(\sigma\sqrt{2}|\Delta s|)^{N+1}/\sqrt{(N+1)!}$ can be smaller than Taylor's $|\Delta s|^{N+1}/(N+1)!$ due to $\sigma^{N+1}$ suppression, yielding conditional superiority (see Appendix 4).

Table 3: Quantitative comparison on the text-to-image generation task with the FLUX.1-dev model. We also report Hi-ClusCa (ClusCa + HiCache) to highlight plug-and-play generality.

| Method | Latency(s)↓ | Speed↑ | FLOPs(T)↓ | Speed↑ | Image Reward↑ | PSNR↑ | SSIM↑ | LPIPS↓ |
|---|---|---|---|---|---|---|---|---|
| FLUX.1 [dev] - 50 steps[†] | 17.12 | 1.00× | 3719.50 | 1.00× | 0.9872 | ∞ | 1.0000 | 0.0000 |
| FLUX.1 [dev] - 25 steps | 8.77 | 1.95× | 1859.75 | 2.00× | 0.9691 | 29.558 | 0.7327 | 0.3115 |
| FLUX.1 [dev] - 20 steps | 7.07 | 2.42× | 1487.80 | 2.62× | 0.9487 | 29.122 | 0.6983 | 0.3599 |
| FLUX.1 [dev] - 17 steps | 6.10 | 2.81× | 1264.63 | 3.13× | 0.9147 | 28.882 | 0.6767 | 0.3912 |
| Chipmunk [29] (34% steps) | 12.72 | 2.02× | 1505.87 | 2.47× | 0.9936 | – | – | – |
| FORA ($\mathcal{N} = 7$) | 4.22 | 4.08× | 670.44 | 5.55× | 0.7418 | 28.315 | 0.5870 | 0.5409 |
| ToCa ($\mathcal{N} = 10, N = 90\%$) | 7.93 | 2.17× | 714.66 | 5.20× | 0.8384 | 28.761 | 0.6068 | 0.4887 |
| TeaCache ($\ell_1 = 1.0$) | 4.92 | 3.48× | 743.63 | 5.00× | 0.8379 | 28.606 | 0.6360 | 0.4773 |
| DBCache [36] ($\mathcal{F} = 4, \mathcal{B} = 0, \mathcal{W} = 4, \mathcal{MC} = 10$) | 4.08 | 4.39× | 816.65 | 4.56× | 0.8245 | – | – | – |
| ClusCa ($\mathcal{N} = 7$) | 4.87 | 3.52× | 674.21 | 5.52× | 0.9480 | 28.630 | 0.6210 | 0.4560 |
| TaylorSeer ($\mathcal{N} = 7, \mathcal{O} = 2$) | 4.84 | 3.54× | 670.44 | 5.55× | 0.9572 | 28.634 | 0.6237 | 0.4520 |
| **Hi-ClusCa**[‡] ($\mathcal{N} = 7, \mathcal{O} = 2, \sigma = 0.5$) | 4.87 | 3.52× | 674.21 | 5.52× | 0.9840 | **28.940** | 0.6560 | 0.4040 |
| **HiCache** ($\mathcal{N} = 7, \mathcal{O} = 2, \sigma = 0.5$) | 4.84 | 3.54× | 670.44 | 5.55× | **0.9979** | 28.937 | **0.6572** | **0.3982** |
| FORA ($\mathcal{N} = 9$) | 3.90 | 4.42× | 596.07 | 6.24× | 0.5457 | 28.233 | 0.5613 | 0.5860 |
| ToCa ($\mathcal{N} = 12, N = 90\%$) | 7.34 | 2.34× | 644.70 | 5.77× | 0.7155 | 28.575 | 0.5677 | 0.5500 |
| TeaCache ($\ell_1 = 1.2$) | 4.45 | 3.85× | 669.27 | 5.56× | 0.7394 | 28.131 | 0.4744 | 0.6765 |
| DBCache [36] ($\mathcal{F} = 1, \mathcal{B} = 0, \mathcal{W} = 4, \mathcal{MC} = 10$) | 3.56 | 5.04× | 651.90 | 5.72× | 0.8796 | – | – | – |
| ClusCa ($\mathcal{N} = 9$) | 4.53 | 3.78× | 599.93 | 6.20× | 0.8440 | 28.370 | 0.5850 | 0.5170 |
| TaylorSeer ($\mathcal{N} = 9, \mathcal{O} = 2$) | 4.50 | 3.80× | 596.07 | 6.24× | 0.8562 | 28.359 | 0.5882 | 0.5088 |
| **Hi-ClusCa**[‡] ($\mathcal{N} = 9, \mathcal{O} = 2, \sigma = 0.5$) | 4.53 | 3.78× | 599.93 | 6.20× | 0.8860 | 28.630 | 0.6380 | 0.4460 |
| **HiCache** ($\mathcal{N} = 9, \mathcal{O} = 2, \sigma = 0.5$) | 4.50 | 3.80× | 596.07 | 6.24× | **0.9113** | **28.647** | **0.6443** | **0.4374** |
| FLUX.1 [schnell][°] - 2 steps | 1.10 | 15.59× | 148.78 | 25.00× | 0.9702 | 28.053 | 0.4720 | 0.6743 |
| FLUX.1 [schnell] - 4 steps | 2.20 | 7.79× | 297.56 | 12.50× | 0.9531 | 28.050 | **0.4731** | **0.6708** |
| **HiCache+ FLUX.1 [schnell]** ($\mathcal{O} = 1, \sigma = 0.3$)* | 1.10 | 15.58× | 148.81 | 24.99× | **0.9975** | **28.057** | 0.4507 | 0.6757 |

- [†] denotes the baseline (FLUX.1 [dev] - 50 steps). Best results are in **bold** and second best are underlined.
- [‡] Hi-ClusCa: ClusCa with its Taylor predictor replaced by HiCache's Hermite predictor.
- [°] FLUX.1 [schnell] is a distilled variant of FLUX.1.
- * HiCache+ FLUX.1 [schnell]: Two full computation steps followed by two HiCache-accelerated steps.

Table 1: **Comparison on text-to-video generation** with HunyuanVideo on VBench.

| Method | Lat.(s)↓ | Spd.↑ | FLOPs(T)↓ | Spd.↑ | VBench(%)↑ |
|---|---|---|---|---|---|
| Original (50 steps) | 145.00 | 1.00× | 29773 | 1.00× | 80.66 |
| 22% steps | 31.87 | 4.55× | 6550 | 4.55× | 78.74 |
| TeaCache | 30.49 | 4.76× | 6550 | 4.55× | 79.36 |
| FORA | 34.39 | 4.22× | 5960 | 5.00× | 78.83 |
| ToCa | 38.52 | 3.76× | 7006 | 4.25× | 78.86 |
| DuCa | 31.69 | 4.58× | 6483 | 4.62× | 78.72 |
| TaylorSeer($\mathcal{N} = 6, \mathcal{O} = 1$) | 31.69 | 4.58× | 5359 | 5.56× | 79.78 |
| **HiCache**($\mathcal{N} = 6, \mathcal{O} = 1$) | 31.71 | 4.58× | 5359 | 5.56× | **79.89** |
| TeaCache | 26.61 | 5.45× | 5359 | 5.56× | 78.32 |
| TaylorSeer($\mathcal{N} = 7, \mathcal{O} = 1$) | 28.82 | 5.03× | 4795 | 6.21× | 79.40 |
| **HiCache**($\mathcal{N} = 7, \mathcal{O} = 1$) | 28.83 | 5.03× | 4795 | 6.21× | **79.51** |
| TaylorSeer($\mathcal{N} = 7, \mathcal{O} = 2$) | 28.83 | 5.03× | 4795 | 6.21× | 79.28 |
| **HiCache**($\mathcal{N} = 7, \mathcal{O} = 2$) | 28.84 | 5.03× | 4795 | 6.21× | **79.65** |

Table 2: **Ablation study with FLUX.1-dev.** $\sigma = 1.0$ means no contraction factor is applied. Hi-Taylor indicates TaylorSeer upgraded with HiCache.

| Method | Speed↑ | Image Reward↑ | PSNR↑ | SSIM↑ | LPIPS↓ |
|---|---|---|---|---|---|
| HiCache ($N = 7, \sigma=0.4$) | 5.55× | 0.9683 | 29.058 | 0.6612 | 0.3914 |
| HiCache ($N = 7, \sigma=0.5$) | 5.55× | **0.9979** | 28.937 | 0.6572 | 0.3982 |
| HiCache ($N = 7, \sigma=0.7$) | 5.55× | 0.9623 | 28.651 | 0.6268 | 0.4479 |
| HiCache ($N = 7, \sigma=1.0$) | 5.55× | 0.7586 | 28.087 | 0.3682 | 0.7208 |
| Hi-Taylor ($N = 7, \sigma=0.5$)[†] | 5.55× | **0.9624** | 29.080 | 0.6571 | 0.3998 |
| TaylorSeer ($N = 7$) | 5.55× | 0.9572 | 28.634 | 0.6237 | 0.4520 |
| FORA ($N = 7$) | 5.55× | 0.7418 | 28.315 | 0.5871 | 0.5409 |
| ToCa ($N = 13, N = 95\%$) | 5.77× | 0.7155 | 28.583 | 0.5677 | 0.5498 |
| DuCa ($N = 12, N = 95\%$) | 6.13× | 0.8382 | 28.947 | 0.5957 | 0.4935 |

# 4 EXPERIMENTS

## 4.1 EXPERIMENT SETTINGS

We conduct comprehensive experiments across four representative generative tasks using state-of-the-art diffusion models: text-to-image generation with FLUX.1-dev (Esser et al., 2024), text-to-video generation with HunyuanVideo (Kong et al., 2024), class-conditional image generation with DiT-XL/2 (Peebles & Xie, 2023), and image super-resolution with a modified Inf-DiT (Yang et al., 2024). Each task employs standard evaluation protocols and widely-adopted benchmarks to ensure fair comparison with existing methods. More detailed setup is provided in Appendix 4.

## 4.2 EXPERIMENT RESULTS

**Text-to-Image Generation.** Table 3 shows HiCache maintains superior quality at high acceleration ratios. At $5.55\times$ speedup ($\mathcal{N} = 7, \mathcal{O} = 2$), its ImageReward (0.9979) exceeds both competitors and the unaccelerated baseline (0.9872). When integrated with ClusCa by (Zheng et al., 2025), HiCache substantially improves ImageReward: from 0.9480 to 0.9840 ($\mathcal{N} = 7$) and from 0.8440 to 0.8860 ($\mathcal{N} = 9$). Figures 4 and 5(a) demonstrate HiCache's visual advantages. As shown in Figure 5(b), HiCache maintains stability across acceleration factors from $1.00\times$ to $9.00\times$, while other feature caching methods exhibit significant degradation in high acceleration ratios.

**Text-to-Video Generation.** Table 3.3.2 shows HiCache outperforms recent methods. At higher orders ($\mathcal{N} = 7, \mathcal{O} = 2$), HiCache's advantage over TaylorSeer becomes more pronounced, validating

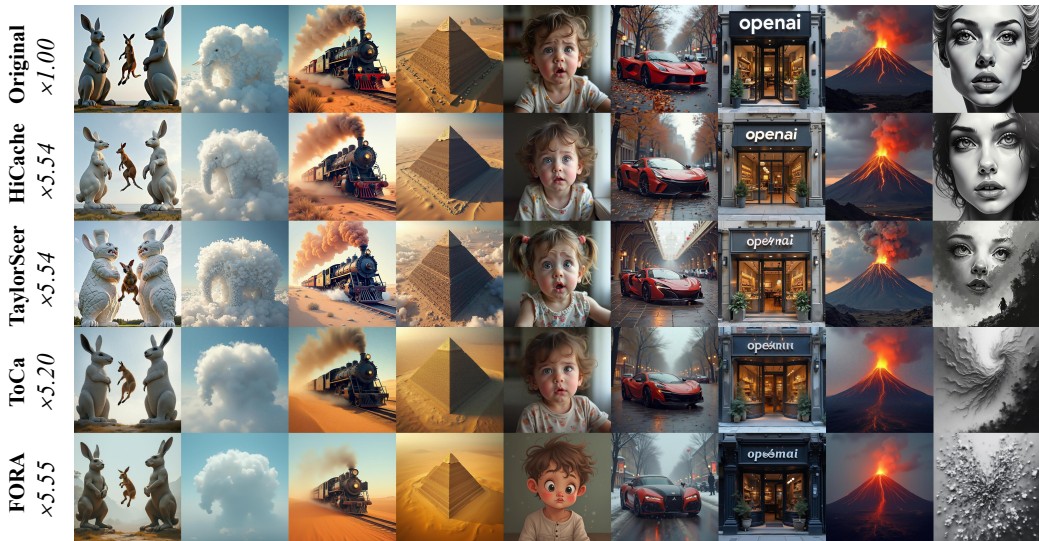

Figure 4: Qualitative comparison on the text-to-image task across diverse prompts.

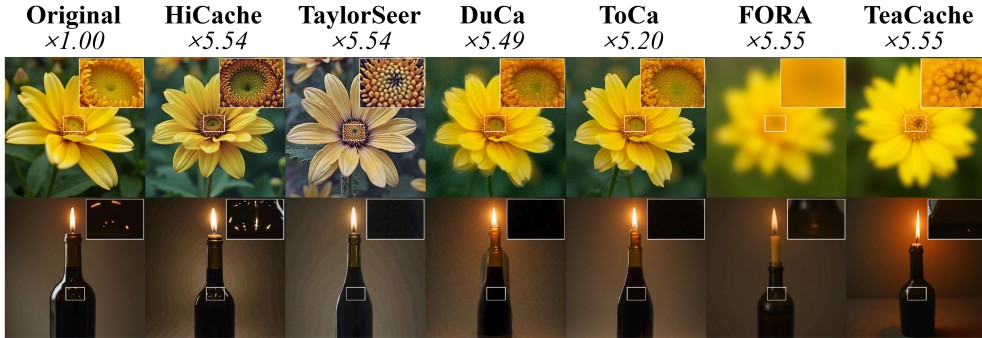

(a) High-frequency detail preservation.

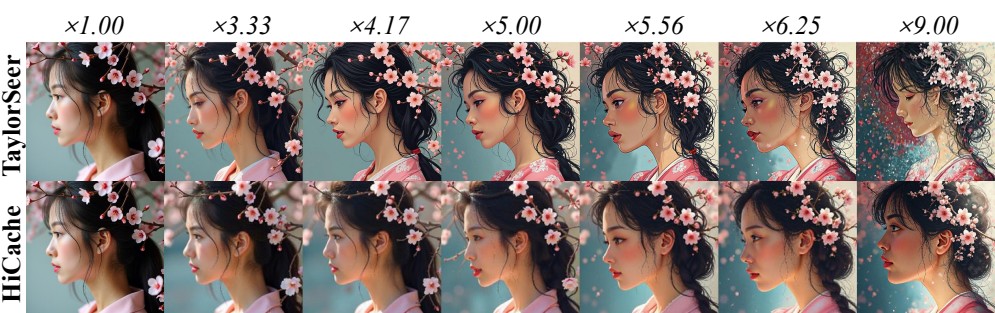

(b) Consistent style and clean backgrounds.

Figure 5: **Detail retention and style consistency.** (a) Superior detail retention. (b) Greater stability than TaylorSeer under higher acceleration, with consistent style and clean outputs.

our theoretical analysis. Figure 6 shows HiCache's superior temporal consistency compared to methods in terms of both temporal consistency and frame generation.

**Class-Conditional Image Generation** Table 4 shows that HiCache outperforms recent accelerators on ImageNet with DiT-XL/2 across speedups, with larger relative gains compared with TaylorSeer at higher acceleration—about $3.2\%$ at $\sim 5.0\times$, $4.5\%$ sFID at $\sim 6.2\times$, and $5.9\%/6.0\%$ FID/sFID at $\sim 7.1\times$—while remaining close to the unaccelerated baseline under aggressive acceleration.

**Super-Resolution.** HiCache achieves $\sim 5.93\times$ theoretical and $\sim 2.43\times$ wall-clock speedup on image super-resolution with Inf-DiT (Yang et al., 2024), following NTIRE 2025 protocol (Chen et al., 2025). Figure 7 shows consistent improvements over TaylorSeer in both PSNR and SSIM. At this

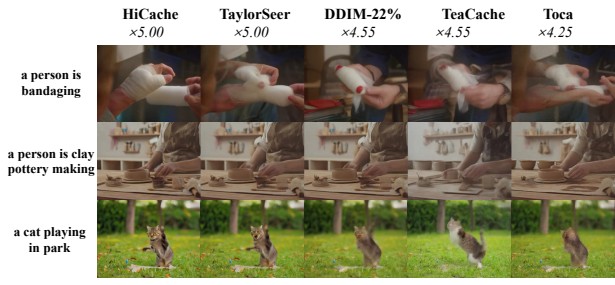

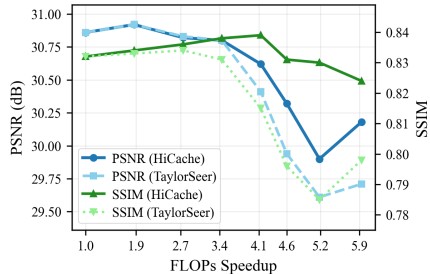

Figure 6: Qualitative comparison of text-to-video generation methods under significant acceleration.

Figure 7: Comparison of HiCache and TaylorSeer on image super-resolution.

Table 4: Quantitative comparison on class-to-image generation on ImageNet with DiT-XL/2.

| Method | Latency(s) ↓ | FLOPs(T) ↓ | Speed ↑ | FID ↓ | sFID ↓ | Inception ↑ |
|---|---|---|---|---|---|---|
| DDIM-50 steps[†] | 0.995 | 23.74 | 1.00× | 2.32 | 4.32 | **241.25** |
| DDIM-20 steps | 0.406 | 9.49 | 2.50× | 3.81 | 5.15 | 221.43 |
| DDIM-8 steps | 0.189 | 3.80 | 6.25× | 23.13 | 19.23 | 120.58 |
| $\Delta$-DiT [5] ($\mathcal{N} = 3$) | 0.173 | 16.14 | 1.47× | 3.75 | — | 207.57 |
| L2C [19] (NFE = 30) | 0.281 | 11.55 | 2.05× | 2.61 | — | 237.83 |
| SmoothCache [17] ($\alpha = 0.22$) | 0.251 | 8.57 | 2.77× | 4.15 | — | 231.71 |
| FORA ($\mathcal{N} = 6$) | 0.427 | 5.24 | 4.98× | 9.24 | 14.84 | 171.33 |
| ToCa ($\mathcal{N} = 9, \mathcal{N} = 95\%$) | 1.016 | 6.34 | 3.75× | 6.55 | 7.10 | 189.53 |
| TaylorSeer ($\mathcal{N} = 6, \mathcal{O} = 4$) | 0.210 | 4.76 | 4.98× | 3.11 | 6.35 | **223.85** |
| **HiCache** ($\mathcal{N} = 6, \mathcal{O} = 4, \sigma = 0.75$) | 0.193 | 4.76 | 4.99× | **3.01** | **6.32** | 223.68 |
| FORA ($\mathcal{N} = 7$) | 0.405 | 3.82 | 6.22× | 12.55 | 18.63 | 148.44 |
| ToCa ($\mathcal{N} = 13, \mathcal{N} = 95\%$) | 1.051 | 4.03 | 5.90× | 21.24 | 19.93 | 116.08 |
| TaylorSeer ($\mathcal{N} = 8, \mathcal{O} = 4$) | 0.190 | 3.82 | 6.22× | 4.40 | 7.34 | **205.00** |
| **HiCache** ($\mathcal{N} = 8, \mathcal{O} = 4, \sigma = 0.75$) | 0.172 | 3.82 | 6.21× | **4.33** | **7.01** | **205.00** |
| FORA ($\mathcal{N} = 8, \mathcal{N} = 95\%$) | 0.405 | 3.34 | 7.10× | 15.31 | 21.91 | 136.21 |
| ToCa ($\mathcal{N} = 13, \mathcal{N} = 98\%$) | 1.033 | 3.66 | 6.48× | 22.18 | 20.68 | 110.91 |
| TaylorSeer ($\mathcal{N} = 9, \mathcal{O} = 4$) | 0.179 | 3.34 | 7.10× | 4.58 | 7.30 | 201.12 |
| **HiCache** ($\mathcal{N} = 9, \mathcal{O} = 4, \sigma = 0.75$) | 0.161 | 3.34 | 7.11× | **4.31** | **6.86** | **203.49** |

● [†] denotes the baseline (DDIM-50 steps). Best results in each block are in **bold** and second best are underlined.

acceleration, the restoration quality degrades only mildly: PSNR drops by $\approx 2.24\%$ (from 30.87 to 30.18) and SSIM by $\approx 1.44\%$ (from 0.832 to 0.820) relative to the interval= 1 baseline. Detailed latency/FLOPs and restoration metrics for each interval are provided in Table 7.

### 4.3 ABLATION STUDY

Table 2 dissects our contributions. Without scaling ($\sigma = 1.0$), HiCache underperforms TaylorSeer due to numerical instability. With optimal scaling ($\sigma = 0.5$), performance improves dramatically. The *Hi-Taylor* variant (TaylorSeer with HiCache's dual-scaling) validates that both our scaling mechanism and Hermite basis contribute independently to HiCache's superiority.

## 5 DISCUSSION

**Empirical Validation of Proposition 2** We validate the Gaussianity of finite differences posited in Proposition 2. Using energy tests (Székely & Rizzo, 2005) across 1st to 5th order feature differences in five key FLUX modules (Esser et al., 2024), the tests confirmed Gaussianity with maximum confidence (p-value = 1.0) across all 25 configurations, providing strong empirical support for our theoretical framework and, in turn, Corollary 1 which motivates Hermite polynomials for Gaussian-correlated processes.

The appendix provides theoretical support through two complementary mechanisms: (i) local linearization yielding conditional Gaussianity for small steps, with residual controlled by second-order bounds, and (ii) aggregation effects invoking multivariate CLT with Berry-Esseen convergence rates.

These mechanisms align with transformer architectures' residual connections and normalization layers that maintain required regularity. Moreover, the robustness analysis shows Hermite bases retain advantages even under approximate Gaussianity, with efficiency degrading at most $O(\epsilon)$ for $\epsilon$-perturbations from exact Gaussian (Appendix 4, Proposition 8). This theoretical foundation, combined with empirical energy test validation, justifies modeling finite differences as approximately Gaussian and motivates the Hermite basis choice in Corollary 1. To further address key concerns beyond the main-page budget, Appendix 4 provides targeted evidence on three fronts: validity of the Gaussian finite-difference assumption (Appendix Table 8), stability under aggressive scaling via adaptive per-layer $\sigma$ (Table 5), and transfer/complementarity across backbones and accelerators (Table 6 and Appendix Table 9). Additional qualitative trajectory evidence is provided in Appendix 4.

**Prediction Simulation Framework** To fairly evaluate HiCache against Taylor expansion, we established a simulation framework using real feature trajectories from the FLUX model with activation interval $N_{\text{interval}} = 6$, where derivative approximations are computed using sparsely sampled data from previous activation steps. Our simulation eliminates cumulative error by initializing each prediction task with ground-truth history. This isolates cascading inaccuracies and enables fair quantitative comparison of Hermite and Taylor bases.

**Empirical Verification of Oscillatory Advantages** Appendix Figure 9 provides empirical evidence for the theoretical advantages of Hermite polynomials' oscillatory properties (discussed in Section 3.3.2). Using the non-cumulative error framework on representative FLUX features, we observe that while both methods perform similarly at order 1, Taylor expansion increasingly over-extrapolates at higher orders, especially at trajectory turning points—precisely confirming our intuition about monomial limitations. HiCache's oscillatory Hermite basis, stabilized by the contraction factor $\sigma$ (Definition 2), maintains accurate fitting across all orders (see Appendix 4 for details).

**Quantitative Validation of Error Bounds** To empirically verify the tighter error bounds discussed in our methodology, we quantitatively compared HiCache against Taylor expansion using the simulation framework. We define the relative error ratio as $R = {}^{\text{Taylor MSE}}/_{\text{Hermite MSE}}$ (where $R > 1$ indicates superior HiCache performance) and analyzed 25 configurations (5 FLUX modules x 5 polynomial orders); Figure 8 summarizes the cumulative error ratios.

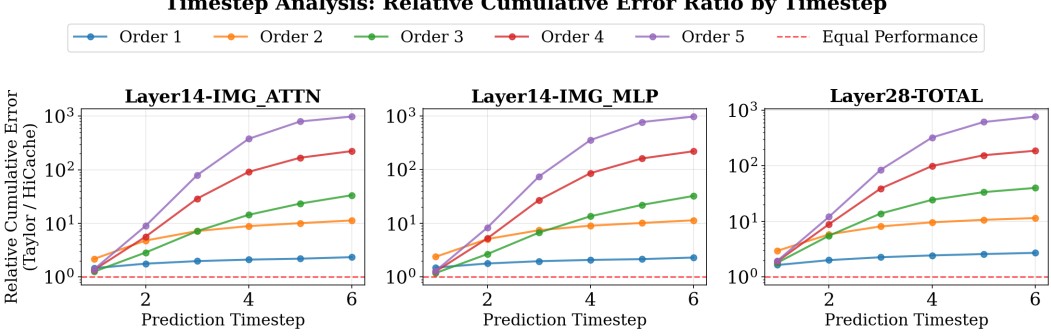

Figure 8: Cumulative error ratios (Taylor MSE / HiCache MSE) across 5 FLUX modules.

The results empirically confirm HiCache's consistent advantage ($R > 1.0$ across all modules), with the performance gap widening for larger prediction steps and higher polynomial orders. These results confirm HiCache's consistent advantage in our parameter regime. In particular, when the contraction condition $\sigma\sqrt{2} \cdot |\Delta s| < 1$ holds and for moderate $N$, the empirical truncation behavior aligns with our conditional analysis in Section 3.

**Additional Evidence on Stability and Generality** To further address key concerns beyond the main-page budget, we summarize three supplementary findings. First, energy-distance Gaussianity tests on FLUX finite differences show strong support across depth and diffusion phases: 75/75 tests pass over all timesteps, 75/75 in the early high-noise phase, 73/75 in the mid transition phase, and 75/75 in the late low-noise phase (Appendix Table 8). This indicates that the Gaussian-motivated Hermite basis is not tied to a narrow operating regime.

Second, adaptive per-layer scaling confirms the practical value of our stability condition. Under interval $= 7$ and order $= 2$, fixed $\sigma = 1.0$ is unstable (SSIM 0.362, LPIPS 0.725, ImageReward 0.736), whereas adaptive per-layer $\sigma$ at comparable initial scale recovers strong quality (SSIM 0.648, LPIPS 0.415, ImageReward 0.957), close to the robust fixed-$\sigma \approx 0.5$ regime (Table 5).

Table 5: Adaptive per-layer $\sigma$ (HiCache-Adaptive) on FLUX.1-dev under interval $= 7$, order $= 2$ (200 prompts, 1024×1024). Here $\sigma_{\text{fixed/ max}}$ denotes the global $\sigma$ for fixed HiCache and $\sigma_{\max}$ for HiCache-Adaptive.

| Method | $\alpha$ | $\sigma_{\text{fixed/ max}}$ | Init. $\sigma$ | CLIP↑ | PSNR↑ | SSIM↑ | LPIPS↓ | ImageReward↑ |
|---|---|---|---|---|---|---|---|---|
| **Baseline** | | | | | | | | |
| TaylorSeer | – | – | – | 27.41 | 28.63 | 0.621 | 0.456 | 0.951 |
| **Comparison at $\sigma \approx 1.0$** | | | | | | | | |
| HiCache-Fixed | – | 1.0 | – | 27.19 | 28.10 | 0.362 | 0.725 | 0.736 |
| HiCache-Adaptive | 1.40 | 1.0 | $\approx 1.0$ | 27.80 | 29.14 | **0.648** | **0.415** | **0.957** |
| **Comparison at $\sigma \approx 0.5$** | | | | | | | | |
| HiCache-Fixed | – | 0.5 | – | 27.62 | 28.93 | 0.655 | 0.404 | 0.974 |
| HiCache-Adaptive | 0.70 | 0.7 | $\approx 0.5$ | 27.87 | 29.11 | 0.646 | 0.415 | 0.971 |

Third, cross-backbone and accelerator results show both transfer and complementarity. On Qwen-Image at higher intervals (e.g., $\mathcal{N} = 8$), HiCache improves ImageReward from $0.750$ to $0.844$ and reduces LPIPS from $0.591$ to $0.531$ versus TaylorSeer (Table 6).

Table 6: Qwen-Image results (1328×1328, 50 steps, 200 prompts) comparing TaylorSeer and HiCache at different intervals, with approximate FLOPs speedup.

| Interval | Mode | FLOPs× | CLIP↑ | ImageReward↑ | PSNR↑ | SSIM↑ | LPIPS↓ |
|---|---|---|---|---|---|---|---|
| 3 | Taylor | 2.78 | 28.96 | 1.214 | 30.44 | 0.800 | 0.208 |
| | HiCache | 2.78 | 28.99 | 1.216 | 30.92 | 0.795 | 0.198 |
| 6 | Taylor | 5.00 | 28.09 | 1.012 | 28.52 | 0.601 | 0.481 |
| | HiCache | 5.00 | 28.62 | 1.070 | 28.72 | 0.613 | 0.422 |
| 7 | Taylor | 5.56 | 27.73 | 0.895 | 28.34 | 0.564 | 0.538 |
| | HiCache | 5.56 | 27.87 | **0.944** | 28.50 | 0.579 | **0.478** |
| 8 | Taylor | 6.25 | 27.01 | 0.750 | 28.23 | 0.519 | 0.591 |
| | HiCache | 6.25 | 27.72 | **0.844** | 28.37 | 0.539 | 0.531 |

On Chipmunk-Flux, HiCache improves ImageReward from $0.845$ to $0.938$ over the Taylor variant at similar latency (3.5s vs. 3.3s per image) (Appendix Table 9). Detailed supplementary analyses are provided in Appendix 4, and additional qualitative trajectory evidence is provided in Appendix 4.

## 6 CONCLUSION

We introduce HiCache, a training-free acceleration strategy that overcomes the core limits of Taylor-series feature caching in diffusion models. Grounded in the empirical observation that feature derivatives follow multivariate Gaussian statistics, we replace monomials with Hermite polynomials—the Karhunen–Loève–optimal basis under Gaussian correlation—and pair them with a simple dual-scaling mechanism that stabilizes coefficients while retaining the oscillatory behavior needed to model non-monotonic trajectories. Across text-to-image, text-to-video, and super-resolution benchmarks, HiCache consistently matches or surpasses prior methods, delivering large speedups without visible quality loss. Because it only swaps the predictive basis, HiCache drops into existing cache-then-forecast pipelines and requires no retraining or architectural changes, offering a principled and practical path to fast diffusion sampling.

ETHICS STATEMENT

This work does not introduce new data collection or involve human subjects. All datasets used are publicly available under their original licenses; no personally identifiable or sensitive information is processed. Generated content may reflect biases present in the underlying datasets; we report standard evaluation metrics and release code to facilitate auditing and reproducibility. By reducing the number of expensive model evaluations, HiCache lowers the compute and energy consumption relative to baselines. We discourage misuse of accelerated generative models for deceptive or harmful purposes and will release our implementation with appropriate usage guidelines.

REPRODUCIBILITY CHECKLIST

**1. General Paper Structure**

1.1. Includes a conceptual outline and/or pseudocode description of AI methods introduced (yes/partial/no/NA) yes

1.2. Clearly delineates statements that are opinions, hypothesis, and speculation from objective facts and results (yes/no) yes

1.3. Provides well-marked pedagogical references for less-familiar readers to gain background necessary to replicate the paper (yes/no) yes

**2. Theoretical Contributions**

2.1. Does this paper make theoretical contributions? (yes/no) yes

If yes, please address the following points:

2.2. All assumptions and restrictions are stated clearly and formally (yes/partial/no) yes

2.3. All novel claims are stated formally (e.g., in theorem statements) (yes/partial/no) yes

2.4. Proofs of all novel claims are included (yes/partial/no) yes

2.5. Proof sketches or intuitions are given for complex and/or novel results (yes/partial/no) yes

2.6. Appropriate citations to theoretical tools used are given (yes/partial/no) yes

2.7. All theoretical claims are demonstrated empirically to hold (yes/partial/no/NA) yes

2.8. All experimental code used to eliminate or disprove claims is included (yes/no/NA) yes

**3. Dataset Usage**

3.1. Does this paper rely on one or more datasets? (yes/no) yes

If yes, please address the following points:

3.2. A motivation is given for why the experiments are conducted on the selected datasets (yes/partial/no/NA) yes

3.3. All novel datasets introduced in this paper are included in a data appendix (yes/partial/no/NA) NA

3.4. All novel datasets introduced in this paper will be made publicly available upon publication of the paper with a license that allows free usage for research purposes (yes/partial/no/NA) NA

3.5. All datasets drawn from the existing literature (potentially including authors' own previously published work) are accompanied by appropriate citations (yes/no/NA) yes

3.6. All datasets drawn from the existing literature (potentially including authors' own previously published work) are publicly available (yes/partial/no/NA) yes

3.7. All datasets that are not publicly available are described in detail, with explanation why publicly available alternatives are not scientifically satisficing (yes/partial/no/NA) NA

## 4. Computational Experiments

4.1. Does this paper include computational experiments? (yes/no) yes

If yes, please address the following points:

4.2. This paper states the number and range of values tried per (hyper-) parameter during development of the paper, along with the criterion used for selecting the final parameter setting (yes/partial/no/NA) yes

4.3. Any code required for pre-processing data is included in the appendix (yes/partial/no) yes

4.4. All source code required for conducting and analyzing the experiments is included in a code appendix (yes/partial/no) yes

4.5. All source code required for conducting and analyzing the experiments will be made publicly available upon publication of the paper with a license that allows free usage for research purposes (yes/partial/no) yes

4.6. All source code implementing new methods have comments detailing the implementation, with references to the paper where each step comes from (yes/partial/no) yes

4.7. If an algorithm depends on randomness, then the method used for setting seeds is described in a way sufficient to allow replication of results (yes/partial/no/NA) yes

4.8. This paper specifies the computing infrastructure used for running experiments (hardware and software), including GPU/CPU models; amount of memory; operating system; names and versions of relevant software libraries and frameworks (yes/partial/no) yes

4.9. This paper formally describes evaluation metrics used and explains the motivation for choosing these metrics (yes/partial/no) yes

4.10. This paper states the number of algorithm runs used to compute each reported result (yes/no) yes

4.11. Analysis of experiments goes beyond single-dimensional summaries of performance (e.g., average; median) to include measures of variation, confidence, or other distributional information (yes/no) yes

4.12. The significance of any improvement or decrease in performance is judged using appropriate statistical tests (e.g., Wilcoxon signed-rank) (yes/partial/no) yes

4.13. This paper lists all final (hyper-)parameters used for each model/algorithm in the paper's experiments (yes/partial/no/NA) yes

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

## AI ASSISTANCE DISCLOSURE

In the preparation of this manuscript, the authors utilized Large Language Models (LLMs) to assist with LaTeX formatting and writing refinement. Specifically, LLMs were employed to help with document structure optimization, mathematical notation consistency, and language polishing to improve clarity and readability. All technical content, experimental results, theoretical contributions, and scientific claims remain entirely the work of the human authors. The use of AI tools was limited to formatting assistance and stylistic improvements, without generating any substantive scientific content or analysis.

## HERMITE POLYNOMIALS: MATHEMATICAL FOUNDATIONS

We adopt the physicist's Hermite polynomials $H_n(x)$, whose standard differential form (Rodrigues formula) is:

$$H_n(x) = (-1)^n e^{x^2} \frac{d^n}{dx^n} e^{-x^2} \tag{7}$$

The Gaussian kernel $e^{-x^2}$ in this formula directly embodies the connection with Gaussian measure. From this differential form, we can derive the recurrence relation:

$$H_{n+1}(x) = 2x H_n(x) - 2n H_{n-1}(x) \tag{8}$$

where $H_0(x) = 1$ and $H_1(x) = 2x$. The first few polynomials are:

$$\begin{aligned}
H_0(x) &= 1 \\
H_1(x) &= 2x \\
H_2(x) &= 4x^2 - 2 \\
H_3(x) &= 8x^3 - 12x \\
H_4(x) &= 16x^4 - 48x^2 + 12
\end{aligned} \tag{9}$$

## DETAILED ALGORITHM DESCRIPTION

HiCache algorithm operates through two main phases. Central to this is the concept of the derivative approximation, $\Delta^k \mathcal{F}$, which approximates the $k$-th order derivative of a feature $\mathcal{F}$ at a given timestep. As defined in Eq. (1), it is computed recursively from lower-order approximations at previous activation steps.

**Phase 1: Cache Update (at activation steps $t_i$ where $t_i \mod N_{\textbf{interval}} = 0$)**

This phase uses finite differences to compute high-order derivative approximations and updates the cache. The interval between adjacent activation steps is $\Delta t_{\text{hist}} = N_{\text{interval}}$.

---

**Algorithm 2** Cache Update with Derivative Approximations

---

**Require:** Current feature $\mathcal{F}(t_i)$, previous activation step $t_{i-1}$, previous cache $\{\Delta^k \mathcal{F}(t_{i-1})\}_{k=0}^{N_{order}}$, interval $N_{\text{interval}}$
**Ensure:** Updated cache $\{\Delta^k \mathcal{F}(t_i)\}_{k=0}^{N_{order}}$
1: $\Delta t_{hist} \leftarrow t_i - t_{i-1}$ {Usually equals $N_{\text{interval}}$}
2: $NewCache[0] \leftarrow \mathcal{F}(t_i)$ {Zeroth-order is the feature itself}
3: **for** $k = 0$ to $N_{order} - 1$ **do**
4:     {Compute $(k + 1)$-th order derivative approximation}
5:     $\Delta^{k+1} \mathcal{F}(t_i) \leftarrow (NewCache[k] - \Delta^k \mathcal{F}(t_{i-1}))/\Delta t_{hist}$
6:     $NewCache[k + 1] \leftarrow \Delta^{k+1} \mathcal{F}(t_i)$
7: **end for**
8: **return** $NewCache$

---

**Phase 2: Feature Prediction (at non-activation steps $t$ where $t \mod N_{\textbf{interval}} \neq 0$)**

---

**Algorithm 3** Feature Prediction

---

**Require:** Current step $t$, most recent activation step $t_{last}$, cache $\{\Delta^k \mathcal{F}(t_{last})\}_{k=0}^{N_{order}}$, contraction factor $\sigma$
**Ensure:** Predicted feature $\hat{\mathcal{F}}$
  1: $\Delta t \leftarrow t_{last} - t$ {Distance from last activation, $1 \le \Delta t < N_{\text{interval}}$}
  2: $\hat{\mathcal{F}} \leftarrow \Delta^0 \mathcal{F}(t_{last})$ {Initialize with zeroth-order term}
  3: **for** $k = 1$ to $N_{order}$ **do**
  4:    {Compute scaled Hermite polynomial value}
  5:    $h_k \leftarrow \tilde{H}_k(-\Delta t)$ {Or equivalently: $h_k \leftarrow (-1)^k \cdot \tilde{H}_k(\Delta t)$}
  6:    $\hat{\mathcal{F}} \leftarrow \hat{\mathcal{F}} + \frac{h_k}{k!} \cdot \Delta^k \mathcal{F}(t_{last})$ {Accumulate}
  7: **end for**
  8: **return** $\hat{\mathcal{F}}$

---

This phase performs efficient prediction based on cached derivative approximations and scaled Hermite polynomials. The prediction step $\Delta t$ ranges from $[1, N_{\text{interval}} - 1]$.

where Hermite polynomials are efficiently computed through the recurrence relation:

- $H_0(x) = 1$

- $H_1(x) = 2x$

- $H_{k+1}(x) = 2x H_k(x) - 2k H_{k-1}(x)$ for $k \ge 1$

This algorithm design balances computational efficiency with prediction accuracy. Key parameters include:

- **Interval parameter** $N_{\textbf{interval}} \in \mathbb{N}^+$: Controls activation frequency, with larger values providing higher acceleration ratio but potentially increased prediction error

- **Contraction factor** $\sigma \in (0, 1)$: Stabilization parameter that ensures numerical stability of Hermite polynomials within the prediction interval

- **Maximum order** $N_{\textbf{order}} \in \mathbb{N}^+$: Expansion order that balances approximation accuracy and computational complexity, with practical implementations typically using $N_{\text{order}} \le 4$

## COMPLETE ERROR ANALYSIS AND THEORETICAL GUARANTEES

This section provides rigorous proofs for the error bounds stated in Proposition 1 (Error Bounds for HiCache) in the main paper. We assume the feature evolution trajectory $f : [0, T] \to \mathcal{F}$ is sufficiently smooth with bounded derivatives up to order $N_{\text{order}} + 1$.

The total prediction error decomposes into three components:
$$\mathbf{E}_{\text{total}} = \mathbf{E}_{\text{truncation}} + \mathbf{E}_{\text{approximation}} + \mathbf{E}_{\text{numerical}} \tag{10}$$

### TRUNCATION ERROR ANALYSIS

Based on the Karhunen-Loève theorem and validated Gaussian properties, the feature function $f(s)$ can be expanded under the weight function $w(x) = e^{-(\sigma x)^2}$ as:
$$f(s_{\text{last}} + \Delta s) = \sum_{k=0}^{\infty} a_k \frac{\tilde{H}_k(\Delta s)}{\sqrt{2^k k! \sqrt{\pi}}} \tag{11}$$

where the coefficients $a_k = \langle f, \tilde{H}_k \rangle_w$ possess rapid decay properties.

**Lemma 1** (Truncation Error Bound). *The truncation error from using order $N_{order}$ satisfies:*
$$\mathbf{E}_{truncation} \le C_2 \frac{(\sigma \sqrt{2} |\Delta s|)^{N_{order}+1}}{\sqrt{(N_{order}+1)!}} \exp\left(\frac{(\sigma \Delta s)^2}{2}\right) \tag{12}$$

*Proof.* Using the asymptotic bound $|H_k(x)| \leq C_1\sqrt{2^k k!}e^{x^2/2}$ and the scaling properties of $\tilde{H}_k$:

$$\mathbf{E}_{\text{truncation}} = \left| \sum_{k=N_{\text{order}}+1}^{\infty} \frac{1}{k!}\tilde{H}_k(\Delta s)\,\Delta^k f(s_{\text{last}}) \right| \tag{13}$$

$$\leq \sum_{k=N_{\text{order}}+1}^{\infty} \frac{\sigma^k |H_k(\sigma\Delta s)|}{k!} \cdot |\Delta^k f| \tag{14}$$

$$\leq C_2 \frac{(\sigma\sqrt{2}|\Delta s|)^{N_{\text{order}}+1}}{\sqrt{(N_{\text{order}}+1)!}} \exp\left(\frac{(\sigma\Delta s)^2}{2}\right) \tag{15}$$

The contraction factor $\sigma < 1$ ensures convergence. $\qquad\square$

**Remark 1** (Comparison with Taylor Expansion). *The Taylor remainder $\mathbf{E}_{truncation}^{Taylor} = O((\Delta s)^{N+1}/(N+1)!)$ lacks the $\sqrt{(N+1)!}$ denominator and $\sigma^{N+1}$ suppression factor, resulting in looser bounds for non-monotonic trajectories.*

FINITE DIFFERENCE APPROXIMATION ERROR

**Lemma 2** (Finite Difference Approximation Error). *For a sufficiently smooth function $f \in C^{k+1}[0,T]$, the backward difference operator $\Delta^{(k)}$ satisfies:*

$$\|\Delta^k f(t) - f^{(k)}(t)\| = \frac{\Delta t_{hist}}{k+1}\|f^{(k+1)}(\xi)\| + O(\Delta t_{hist}^2) \tag{16}$$

*for some $\xi \in [t - k\Delta t_{hist}, t]$.*

**Lemma 3** (Gaussian Decay in Feature Dynamics). *Under the Gaussian feature dynamics assumption, the expected norm of derivative approximations satisfies:*

$$\mathbb{E}[\|\Delta^k \mathbf{F}\|^2] = \frac{\sigma_{\mathbf{F}}^2}{\Gamma(k/2+1)} \cdot (1 + O(k^{-1})) \tag{17}$$

*where $\sigma_{\mathbf{F}}^2$ is the feature variance and $\Gamma$ is the gamma function.*

**Lemma 4** (Total Approximation Error). *The cumulative finite difference error in HiCache satisfies:*

$$\mathbf{E}_{approximation} \leq C_4\Delta t_{hist}\sqrt{N_{order}} \tag{18}$$

*Proof.* Combining Lemma 2 and Lemma 3:

$$\mathbf{E}_{\text{approximation}} = \sum_{k=1}^{N_{\text{order}}} \mathbb{E}\left[\|\Delta^k f - f^{(k)}\| \cdot \frac{|\tilde{H}_k(\Delta s)|}{k!}\right] \tag{19}$$

$$\leq \sum_{k=1}^{N_{\text{order}}} \frac{C_3\Delta t_{\text{hist}}}{k+1} \cdot \frac{\sigma^k}{\sqrt{\Gamma(k/2+1)}} \tag{20}$$

$$\leq C_4\Delta t_{\text{hist}} \sum_{k=1}^{N_{\text{order}}} \frac{1}{k^{3/2}} \tag{21}$$

$$\leq C_4\Delta t_{\text{hist}}\sqrt{N_{\text{order}}} \tag{22}$$

The rapid decay from both the Gaussian property and factorial normalization ensures convergence. $\qquad\square$

NUMERICAL STABILITY ANALYSIS

The recurrence relation $H_{k+1}(x) = 2xH_k(x) - 2kH_{k-1}(x)$ becomes numerically unstable when $|x| \gg 1$. The contraction factor $\sigma$ limits the effective computation domain to $|\sigma\Delta s| < 2$, maintaining condition number $\kappa \approx O(1)$.

Due to the validated Gaussian properties of feature differences, Hermite coefficients possess optimal decay properties:

$$\text{Var}(c_k) \propto \frac{1}{k!}, \quad \text{Cov}(c_i, c_j) = 0 \text{ for } i \neq j \tag{23}$$

*Remark.* The uncorrelatedness $\text{Cov}(c_i, c_j) = 0$ refers to coefficients obtained via $L^2(\gamma)$ orthogonal projection; it does not imply statistical independence in general.

**Lemma 5** (Numerical Error Bound). *The floating-point error in HiCache is bounded by:*

$$\mathbf{E}_{numerical} \leq C_5 \epsilon_{machine} \sqrt{\sum_{k=0}^{N_{order}} \frac{\sigma^{2k}}{k!}} \leq C_5 \epsilon_{machine} e^{\sigma^2/2} \tag{24}$$

**Lemma 6** (Envelope bound for Hermite and scaled Hermite). *For all $n \in \mathbb{N}$ and $x \in \mathbb{R}$, there exists a universal constant $C > 0$ such that $|H_n(x)| \leq C\, e^{x^2/2}(\sqrt{2}|x| + 1)^n \sqrt{n!}$. Consequently, for $\sigma \in (0, 1)$, the scaled polynomials satisfy*

$$|\tilde{H}_n(x)| = \sigma^n |H_n(\sigma x)| \leq C\, e^{(\sigma x)^2/2}(\sqrt{2}\,\sigma|x| + 1)^n \sqrt{n!}, \tag{25}$$

*exhibiting geometric damping in $n$ controlled by $\sigma$.*

*Proof.* The bound for $H_n$ follows from standard inequalities derived via the generating function

$$e^{-t^2 + 2xt} = \sum_{n \geq 0} H_n(x) \frac{t^n}{n!},$$

and Cauchy's estimates, or from known asymptotics (e.g., Plancherel–Rotach). Multiplying by $\sigma^n$ and contracting the input to $\sigma x$ yields the scaled bound. □

TOTAL ERROR BOUND AND PERFORMANCE ADVANTAGES

**Theorem 1** (Main Error Theorem - Complete Proof of Proposition 4 in the main paper). *Combining Lemma 1, Lemma 4, and Lemma 5, the total HiCache prediction error satisfies:*

$$\|\mathbf{E}_{total}\| \leq \underbrace{C_2 \frac{(\sigma\sqrt{2}|\Delta s|)^{N_{order}+1}}{\sqrt{(N_{order}+1)!}} e^{(\sigma \Delta s)^2/2}}_{\text{Truncation}} + \underbrace{C_4 \Delta t_{hist} \sqrt{N_{order}}}_{\text{Approximation}} + \underbrace{C_5 \epsilon_{machine}}_{\text{Numerical}} \tag{26}$$

**Corollary 2** (Conditional Advantage over Taylor). *Under the condition $\sigma\sqrt{2} \cdot |\Delta s| < 1$ and moderate $N$, HiCache can achieve:*

0.1 **Smaller truncation error:** *The Hermite term $(\sigma\sqrt{2} \cdot |\Delta s|)^{N+1}/\sqrt{(N+1)!}$ can be smaller than Taylor's $|\Delta s|^{N+1}/(N+1)!$ due to the exponential suppression factor $\sigma^{N+1}$*

0.2 **Oscillatory basis structure:** *That naturally captures non-monotonic feature dynamics*

0.3 **Adaptive basis:** *Hermite polynomials naturally match Gaussian feature statistics*

*Without the condition $\sigma\sqrt{2} \cdot |\Delta s| < 1$, universal faster convergence cannot be guaranteed.*

*Discussion.* The conditional advantage arises from the interplay between the denominators and the contraction factor. While $1/\sqrt{(N+1)!}$ is actually larger than $1/(N+1)!$, the exponential suppression $\sigma^{N+1}$ (with $\sigma < 1$) in the numerator can overcome this difference when $\sigma\sqrt{2} \cdot |\Delta s| < 1$, yielding smaller overall error. □

DETAILED EXPERIMENTAL SETUP

This section provides comprehensive technical details for all experimental configurations mentioned in Section 4.1.

MODEL AND TASK SPECIFICATIONS

**Text-to-Image Generation (FLUX.1-dev):**  We employ the FLUX.1-dev model for high-resolution text-to-image synthesis. Images are generated at 1024x1024 resolution using 200 high-quality prompts sourced from the DrawBench benchmark. Quality assessment is performed using ImageReward, a robust perceptual metric for text-to-image alignment.

**Text-to-Video Generation (Hunyuan-DiT):**  The Hunyuan-DiT model is used for temporal content generation. We generate 4, 730 video clips from 946 diverse prompts, evaluating performance using the comprehensive VBench framework which assesses multiple aspects including temporal consistency, object persistence, and text-video alignment.

**Class-Conditional Image Generation (DiT-XL/2):**  Following standard academic protocols, we conduct experiments on ImageNet using the DiT-XL/2 architecture. We generate 50, 000 samples across all ImageNet classes and report FID-50k, sFID, and Inception Score for comprehensive quality assessment.

**Image Super-Resolution (Inf-DiT):**  We employ a modified patch-based Inf-DiT model for super-resolution tasks. Evaluation is conducted on the DIV8K dataset using standard PSNR and SSIM metrics to assess restoration quality and perceptual fidelity.

HARDWARE AND COMPUTATIONAL RESOURCES

All experiments are conducted on enterprise-grade GPU infrastructure:

- FLUX.1-dev experiments: NVIDIA H800 GPU

- DiT-XL/2 experiments: NVIDIA A800 GPU

- Hunyuan-DiT experiments: NVIDIA H100 GPU

- Inf-DiT super-resolution: NVIDIA A100 GPU

EVALUATION PROTOCOLS

Each task employs task-specific evaluation methodologies:

- **Text-to-Image:** ImageReward scores for semantic alignment and image quality

- **Text-to-Video:** VBench comprehensive evaluation including temporal consistency and motion quality

- **Class-Conditional:** Standard ImageNet metrics (FID-50k, sFID, Inception Score)

- **Super-Resolution:** Pixel-level metrics (PSNR, SSIM) for restoration accuracy

SUPER-RESOLUTION TASK: DETAILED EXPERIMENTAL RESULTS

This section provides comprehensive experimental results for HiCache's evaluation on image super-resolution using the patch-based Inf-DiT model. The experiments were conducted using standard evaluation metrics including PSNR and SSIM for restoration quality assessment.

Table 7 presents detailed quantitative results across different acceleration intervals, comparing HiCache with TaylorSeer across multiple evaluation metrics including PSNR and SSIM.

The results demonstrate that HiCache maintains superior performance across all metrics, with particularly notable advantages at higher acceleration ratios. At interval=8, HiCache achieves a PSNR of 30.18 (only 0.69 drop from baseline), while TaylorSeer degrades to 29.71 (1.17 drop).

Table 7: Performance Comparison of Different Guiders on Image Super-Resolution Task

| Method | Interval | Latency (s) | Time Speedup | FLOPs (G) | FLOPs Speedup | Restoration Track | |
|---|---|---|---|---|---|---|---|
| | | | | | | PSNR↑ | SSIM↑ |
| HiCache | 1 | 256.3 | 1.00× | 12400 | 1.00× | 30.87 | 0.832 |
| | 2 | 174.4 | 1.47× | 6639 | 1.87× | 30.92 | 0.835 |
| | 3 | 141.6 | 1.81× | 4516 | 2.75× | 30.83 | 0.836 |
| | 4 | 127.7 | 2.01× | 3606 | 3.44× | 30.80 | 0.838 |
| | 5 | 118.6 | 2.16× | 3000 | 4.13× | 30.62 | 0.839 |
| | 6 | 113.7 | 2.25× | 2696 | 4.60× | 30.32 | 0.831 |
| | 7 | 108.9 | 2.35× | 2393 | 5.18× | 29.89 | 0.829 |
| | 8 | 105.6 | **2.43×** | 2090 | **5.93×** | **30.18** (-0.69) | **0.820** (-0.012) |
| TaylorSeer | 1 | 259.5 | 1.00× | 12400 | 1.00× | 30.88 | 0.832 |
| | 2 | 172.7 | 1.50× | 6639 | 1.87× | 30.90 | 0.833 |
| | 3 | 139.3 | 1.86× | 4515 | 2.75× | 30.85 | 0.834 |
| | 4 | 125.0 | 2.08× | 3606 | 3.44× | 30.80 | 0.831 |
| | 5 | 115.9 | 2.24× | 2999 | 4.13× | 30.42 | 0.815 |
| | 6 | 110.9 | 2.34× | 2696 | 4.60× | 29.95 | 0.796 |
| | 7 | 105.9 | 2.45× | 2392 | 5.18× | 29.61 | 0.786 |
| | 8 | 101.8 | **2.55×** | 2089 | **5.94×** | **29.71** (-1.17) | **0.799** (-0.033) |

Note: HiCache uses $\sigma = 0.5$. Values in (blue) denote the change relative to the interval=1 baseline.

This represents the first successful application of cache-based acceleration to super-resolution tasks, expanding the applicability of feature caching beyond traditional text-to-image and video generation.

## THEORETICAL ANALYSIS OF GAUSSIAN FEATURE DYNAMICS AND HERMITE OPTIMALITY

This section provides theoretical analysis supporting the core claims in Proposition 2 (HiCache Feature Prediction) in the main paper. We examine both the empirical foundation and theoretical justification for using Hermite polynomials as the prediction basis.

### ON THE GAUSSIANITY OF FEATURE DERIVATIVES

We present two complementary, verifiable mechanisms that yield quantitative Gaussian approximations for finite differences $\Delta \mathbf{F}_t$.

**Proposition 4** (Local linearization $\Rightarrow$ approximate Gaussian). *Let $x_t = \alpha_t x_0 + \sigma_t \varepsilon_t$ with $\varepsilon_t \sim \mathcal{N}(0, I)$ independent of $x_0$. Let $F : \mathbb{R}^d \times \mathbb{R} \to \mathbb{R}^p$ be twice differentiable with bounded second derivatives $\|\nabla_x^2 F\|_{op} \leq L_x, |\partial_{tt} F| \leq L_t, \|\partial_t \nabla_x F\|_{op} \leq L_{xt}$ on a neighborhood. For a small step $\delta > 0$, define $\Delta F_t := F(x_t, t) - F(x_{t-\delta}, t - \delta)$. Then there exist $A_t, b_t$ and residual $R_t$ such that*

$$\Delta F_t = A_t (\varepsilon_t - \tilde{\varepsilon}_{t-\delta}) + b_t + R_t, \qquad \tilde{\varepsilon}_{t-\delta} \sim \mathcal{N}(0, I), \tag{27}$$

*with $\mathbb{E} \|R_t\|_2 \leq C (L_x \mathbb{E}\|x_t - x_{t-\delta}\|_2^2 + L_t \delta^2 + L_{xt} \delta \mathbb{E}\|x_t - x_{t-\delta}\|_2) = O(\delta)$.*

*Proof.* By the two-variable Taylor formula around $(x_t, t)$,

$$F(x_t, t) - F(x_{t-\delta}, t - \delta) = \nabla_x F(x_t, t)(x_t - x_{t-\delta}) - \partial_t F(x_t, t) \delta + R_t, \tag{28}$$

with $\mathbb{E} \|R_t\|_2$ controlled by the stated bounds. Using $x_t - x_{t-\delta} = (\alpha_t - \alpha_{t-\delta})x_0 + \sigma_t \varepsilon_t - \sigma_{t-\delta}\varepsilon_{t-\delta}$ and grouping constants into $b_t$ yields the claimed linear-Gaussian leading term. $\square$

**Proposition 5** (Aggregation $\Rightarrow$ CLT with Berry–Esseen bound). *Suppose $\Delta F_t = \sum_{i=1}^{M} Y_{i,t}$ where $Y_{i,t} \in \mathbb{R}^p$ are zero-mean contributions (e.g., heads/tokens/neurons) with uniformly bounded third*

*moments and a bounded-degree dependency graph. Let $S_t = \sum_i \text{Cov}(Y_{i,t})$ and define $\eta_t :=$ $\max_{\|u\|=1} \frac{\sum_i \mathbb{E}|u^\top Y_{i,t}|^3}{(u^\top S_t u)^{3/2}}$. Then for any unit $u$,*

$$\sup_{x \in \mathbb{R}} \left| \mathbb{P}(u^\top \Delta F_t \le x) - \Phi\left(\frac{x}{\sqrt{u^\top S_t u}}\right) \right| \le C \eta_t, \tag{29}$$

*and $\Delta F_t \Rightarrow \mathcal{N}(0, S_t)$ as $\eta_t \to 0$.*

*Proof.* By the Cramér–Wold device, analyze $u^\top \Delta F_t = \sum_i X_i$ with $X_i := u^\top Y_{i,t}$. A Berry–Esseen bound for bounded-degree dependency graphs controls the Kolmogorov distance by $C \eta_t$. Uniformity over $u$ yields the multivariate statement. $\square$

**Diagnostics.** We monitor Gaussianity via energy tests, dispersion proxies for $\eta_t$ (e.g., effective number of contributors), and local linearization error versus $\delta$.

**Central Limit Theorem Perspective** Consider the feature update through a transformer block:

$$\mathbf{F}_{t+1} = \mathbf{F}_t + \text{MHSA}(\mathbf{F}_t) + \text{MLP}(\mathbf{F}_t) \tag{30}$$

The finite difference approximations aggregate many such updates:

$$\Delta^k \mathbf{F}_t = \frac{1}{\Delta t^k} \sum_{i=1}^{k} (-1)^{k-i} \binom{k}{i} \mathbf{F}_{t-i\Delta t} \tag{31}$$

**Proposition 6** (Asymptotic Gaussianity). *Under mild regularity conditions, if the individual feature updates are weakly dependent with finite variance, the normalized finite differences converge in distribution to a Gaussian:*

$$\frac{\Delta^k \mathbf{F}_t - \mathbb{E}[\Delta^k \mathbf{F}_t]}{\sqrt{Var[\Delta^k \mathbf{F}_t]}} \xrightarrow{d} \mathcal{N}(0, I) \tag{32}$$

*as the feature dimension $d \to \infty$.*

*Justification:* This follows from the multivariate central limit theorem for weakly dependent random vectors. The transformer's residual connections and normalization layers help maintain the required regularity conditions.

**Random Matrix Theory Perspective** The attention mechanism involves random-like weight matrices due to the softmax operation on query-key similarities:

$$\text{Attention}(Q, K, V) = \text{softmax}\left(\frac{QK^T}{\sqrt{d_k}}\right) V \tag{33}$$

**Lemma 7** (Gaussian Universality in Random Matrices). *For large transformer models with random initialization and sufficient depth, the distribution of feature differences approaches a universal form that, under appropriate scaling, exhibits Gaussian characteristics in the bulk of the spectrum.*

This is related to the Wigner semicircle law and its generalizations, which predict Gaussian-like behavior in high-dimensional random matrix ensembles.

**Diffusion Process Approximation** In the continuous-time limit, the discrete diffusion process can be approximated by a stochastic differential equation:

$$d\mathbf{F}_t = \mu(\mathbf{F}_t, t)dt + \sigma(\mathbf{F}_t, t)dW_t \tag{34}$$

where $W_t$ is a Wiener process. Under appropriate conditions on the drift $\mu$ and diffusion $\sigma$ coefficients, the marginal distributions of such processes exhibit Gaussian characteristics, especially for the increments.

HERMITE POLYNOMIALS AS OPTIMAL BASIS

Given the empirically validated Gaussian nature, we now establish why Hermite polynomials form an optimal basis:

**Proposition 7** (Projection optimality in $L^2(\gamma)$). *Let $\gamma$ be a Gaussian measure on the horizon variable. For any target offset function $y$ and any degree-$N$ polynomial $p$, the orthogonal projection $P_N y$ of $y$ onto $\mathrm{span}\{H_0, \ldots, H_N\}$ satisfies*

$$\|y - P_N y\|_{2,\gamma} \leq \|y - p\|_{2,\gamma}. \tag{35}$$

*Thus truncated Hermite expansion minimizes the Gaussian-weighted L2 error within degree-$N$ polynomials.*

*Proof.* Hermite polynomials form an orthogonal basis of $L^2(\gamma)$. For any closed subspace $\mathcal{V}_N := \mathrm{span}\{H_0, \ldots, H_N\}$, the orthogonal projection $P_N$ minimizes distance to $\mathcal{V}_N$ by the Pythagorean theorem: for any $v \in \mathcal{V}_N$, $\|y - P_N y\|_{2,\gamma} \leq \|y - v\|_{2,\gamma}$. Choosing $v = p$ gives the claim. $\square$

**Theorem 2** (Karhunen-Loève Expansion for Gaussian Processes). *Let $X(t)$ be a zero-mean Gaussian process with covariance kernel $K(s,t)$. If $K$ has the form:*

$$K(s,t) = \exp\left(-\frac{(s-t)^2}{2\tau^2}\right) \tag{36}$$

*then the eigenfunctions of the covariance operator are Hermite functions, and the optimal $L^2$ expansion is:*

$$X(t) = \sum_{n=0}^{\infty} \xi_n \psi_n(t) \tag{37}$$

*where $\psi_n$ are scaled Hermite functions and $\xi_n$ are uncorrelated Gaussian coefficients.*

*Proof sketch.* The covariance operator $\mathcal{K}$ defined by:

$$(\mathcal{K}f)(t) = \int K(s,t)f(s)ds \tag{38}$$

has Hermite functions as eigenfunctions when $K$ has Gaussian form. This follows from the fact that Hermite functions are eigenfunctions of the Fourier transform composed with multiplication by a Gaussian, which is essentially what the Gaussian kernel operator represents. $\square$

**Corollary 3** (Optimality for Feature Prediction). *If feature differences $\Delta^k \mathbf{F}$ follow a multivariate Gaussian distribution, then the Hermite polynomial basis minimizes the expected squared prediction error:*

$$\mathbb{E}\left[\left\|\mathbf{F}_{t-\Delta t} - \sum_{k=0}^{N} c_k \phi_k(\Delta t)\right\|^2\right] \tag{39}$$

*among all orthogonal polynomial bases $\{\phi_k\}$.*

*Proof sketch.* By the Karhunen-Loève theorem, the optimal basis for representing a stochastic process is given by the eigenfunctions of its covariance operator. For Gaussian processes with appropriate covariance structure, these are Hermite functions. The optimality follows from the fact that this expansion minimizes the mean squared error for any finite truncation order $N$. $\square$

ROBUSTNESS UNDER APPROXIMATE GAUSSIANITY

Even if the Gaussianity assumption is only approximately satisfied, Hermite polynomials maintain advantages:

**Proposition 8** (Robustness to Non-Gaussianity). *Let $p(\mathbf{x})$ be a distribution with finite moments that can be expressed as:*

$$p(\mathbf{x}) = \phi(\mathbf{x})(1 + \epsilon h(\mathbf{x})) \tag{40}$$

*where $\phi(\mathbf{x})$ is the Gaussian density, $|h(\mathbf{x})| \leq C$, and $\epsilon$ is small. Then the relative efficiency of Hermite basis compared to monomials degrades at most as $O(\epsilon)$.*

This robustness property ensures that even under model misspecification, Hermite polynomials retain their advantages over standard polynomial bases.

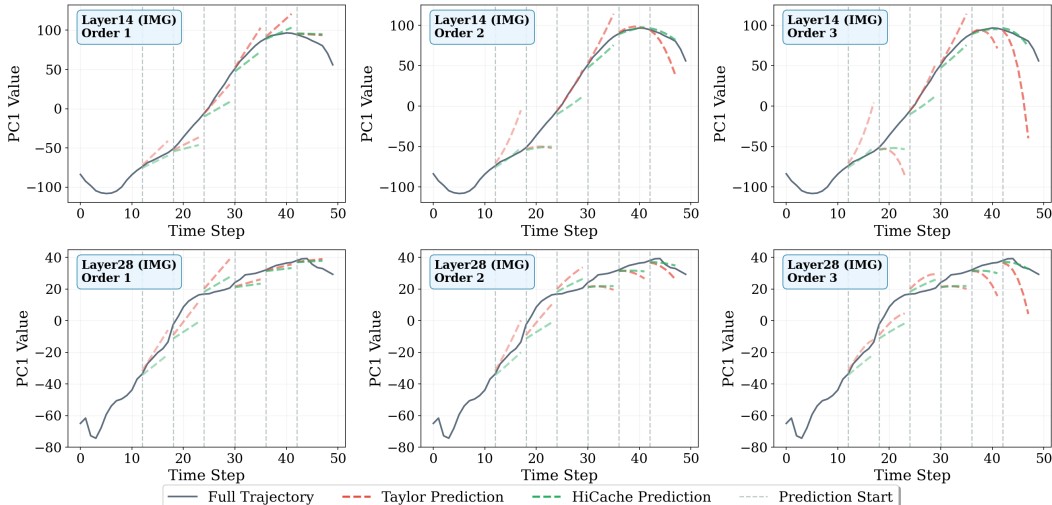

Figure 9: Visualization example of trajectory prediction for different polynomial orders under FLUX architecture. HiCache demonstrates more stable fitting at trajectory turning points, while Taylor expansion over-extrapolates.

## SUMMARY AND PRACTICAL IMPLICATIONS

While we cannot provide a complete first-principles proof that neural network features must exhibit Gaussian dynamics, we have shown:

0.1 **Theoretical Support**: Multiple theoretical frameworks (CLT, random matrix theory, diffusion approximations) suggest Gaussianity is a reasonable expectation in high-dimensional neural networks.

0.2 **Conditional Optimality**: Given the empirically validated Gaussian nature, the optimality of Hermite polynomials follows rigorously from the Karhunen-Loève theorem.

0.3 **Robustness**: Even under approximate Gaussianity, Hermite bases maintain advantages over monomials due to their oscillatory nature and better numerical conditioning.

These theoretical insights, combined with our extensive empirical validation (Section 5.1), provide strong justification for the HiCache design choice of using Hermite polynomials as the prediction basis.

## ADDITIONAL QUALITATIVE RESULTS

### EMPIRICAL VERIFICATION OF OSCILLATORY ADVANTAGES

Figure 9 provides empirical evidence for the theoretical advantages of Hermite polynomials' oscillatory properties discussed in the main paper. Using the non-cumulative error framework on representative FLUX features, we observe that while both methods perform similarly at order 1, Taylor expansion increasingly over-extrapolates at higher orders, especially at trajectory turning points—precisely confirming our theoretical analysis about monomial limitations. HiCache's oscillatory Hermite basis, stabilized by the contraction factor $\sigma$, maintains accurate fitting across all orders.

Table 8: Energy-distance Gaussianity tests on FLUX.1-dev finite differences (15 layer–module configurations $\times$ 5 orders). "Pass" denotes p-value $> 0.05$.

| Setting | Configs$\times$Orders | Tests | Pass | Fail | Pass rate |
|---|---|---|---|---|---|
| All timesteps (all layers/modules) | $15 \times 5$ | 75 | 75 | 0 | 100% |
| Early high-noise phase | $15 \times 5$ | 75 | 75 | 0 | 100% |
| Mid transition phase | $15 \times 5$ | 75 | 73 | 2 | 97.3% |
| Late low-noise phase | $15 \times 5$ | 75 | 75 | 0 | 100% |

## ADDITIONAL EXPERIMENTAL RESULTS

This section reports additional experiments that complement the main text: (i) empirical Gaussianity tests for finite differences on FLUX.1-dev, (ii) an adaptive per-layer $\sigma$ scheme motivated by our truncation-error analysis, and (iii) extra backbones and accelerators (Qwen-Image and Chipmunk-Flux).

### EMPIRICAL GAUSSIANITY VIA ENERGY DISTANCE TESTS ON FLUX.1-DEV

To support the Gaussian finite-difference assumption underlying our Hermite basis, we conduct a comprehensive energy-distance test on FLUX.1-dev features. We collect internal trajectories at six representative layers (4/10/14/20/28/42), covering both early dual-stream and late single-stream blocks, and separately track image/text attention and MLP streams when applicable. For each (layer, module) configuration we compute 1st–5th order finite differences over timesteps and apply a high-dimensional energy distance test (Székely & Rizzo) against a multivariate Gaussian fitted by the empirical mean and covariance, using 500 parametric bootstrap simulations and 400 randomly sampled spatial patches.

Using all timesteps, we obtain $15 \times 5 = 75$ tests in total and observe that *all* p-values are above $0.05$. We further split the trajectory into three regimes on the finite-difference time axis: *early high-noise* ([0,15)), *mid transition* ([15,35)), and *late low-noise* ([35,50)), and rerun the tests in each regime (again 75 tests per regime). The aggregated pass rates are summarized in Table 8.

Overall, finite-difference features on FLUX.1-dev are empirically very close to Gaussian across depth, modality, and time: all 75 all-timestep tests and 150/150 tests in the early/late phases pass at the 0.05 level, and only 2/75 high-order shallow-text configurations in the mid-transition phase fall slightly below 0.05. This supports the modeling assumption that motivates our Hermite basis.

### ADAPTIVE PER-LAYER $\sigma$ ON FLUX.1-DEV

Section 3 analyzes Hermite truncation errors and shows that stable extrapolation requires a condition of the form $\sigma\sqrt{2}|\Delta s| \lesssim 1$. Rather than manually tuning a single global $\sigma$ (e.g., $\sigma = 0.5$), we design a training-free adaptive scheme that chooses a per-layer $\sigma_\ell$ based on the observed scale of feature differences, while respecting this stability constraint.

For each layer $\ell$, we track first-order finite differences $\Delta F_{\ell,t}$ and compress them into a scalar magnitude $D_{\ell,t}$ via an $\ell_2$ norm over channels followed by averaging over space and batch. We then maintain an EMA-style RMS estimate

$$q_\ell^{(t)} = \sqrt{(1-\beta)\left(q_\ell^{(t-1)}\right)^2 + \beta\, D_{\ell,t}^2}, \tag{41}$$

initialized at $q_\ell^{(0)} \approx 1$. The per-layer contraction factor is set to

$$\sigma_\ell = \min\left(\sigma_{\max}, \frac{\alpha}{\sqrt{2}\,q_\ell + \varepsilon}\right), \tag{42}$$

where $\alpha > 0$ determines the desired initial scale (e.g., $\sigma_\ell \approx 0.5$ or $\approx 1.0$ when $q_\ell \approx 1$), $\sigma_{\max}$ caps large values for stability, and $\varepsilon > 0$ avoids division by zero. In practice this adaptive rule simply replaces the fixed scale factor in HiCache with $\sigma_\ell$, without adding any learned parameters.

Table 9: Chipmunk-Flux experiments (flux-dev, 1024×1024, 50 steps, 200 prompts). All accelerated configurations use interval = 5, order = 2.

| Mode | CLIP↑ | ImageReward↑ | PSNR↑ | SSIM↑ | LPIPS↓ | Latency (s/img) |
|---|---|---|---|---|---|---|
| Pure (Chipmunk-Flux) | 27.73 | 0.931 | 28.22 | 0.434 | 0.677 | 5.8 |
| + Taylor | 27.34 | 0.845 | 28.12 | 0.380 | 0.746 | 3.3 |
| + HiCache ($\sigma$=0.5) | 27.88 | **0.938** | 28.12 | 0.415 | 0.704 | 3.5 |

We evaluate this adaptive $\sigma$ on FLUX.1-dev (1024×1024, 50 steps, interval = 7, order = 2, 200 prompts) and compare against TaylorSeer and fixed-$\sigma$ HiCache. All configurations share the same acceleration setting (interval = 7, max_order = 2). The quantitative metrics are reported in Table 5 in the main text.

In the moderate regime where fixed $\sigma = 0.5$ already provides strong performance, adaptive $\sigma$ closely matches or slightly improves this baseline (differences in the reported metrics are within 0.02). When $\sigma$ is increased to 1.0, the fixed-$\sigma$ variant becomes unstable (SSIM decreases from $\approx 0.655$ to $\approx 0.362$ and LPIPS increases from $\approx 0.404$ to $\approx 0.725$), whereas adaptive $\sigma$ with initial scale $\approx 1.0$ remains stable and yields quality close to the $\sigma = 0.5$ regime. This supports the theoretical motivation that the adaptive update can automatically prevent overly aggressive $\sigma$ choices while preserving performance when the baseline configuration is already well behaved.

QWEN-IMAGE TEXT-TO-IMAGE EXPERIMENTS

We next evaluate HiCache on Qwen-Image (Wu et al., 2025) to assess whether the conclusions from FLUX transfer to a different text-to-image backbone. All runs use the same diffusion sampler and hyperparameters; the only differences are whether TaylorSeer or HiCache is enabled and the acceleration interval. We report CLIP, ImageReward, PSNR, SSIM, and LPIPS over 200 prompts at 1328×1328 resolution with 50 sampling steps.

Results are summarized in Table 6 in the main text. At interval 3 (mild acceleration) TaylorSeer and HiCache obtain very similar scores. For larger intervals (6–8), HiCache consistently achieves higher CLIP, ImageReward, and lower LPIPS than TaylorSeer, while keeping PSNR and SSIM at a comparable level, indicating that the Hermite-based forecast is more robust under stronger acceleration.

CHIPMUNK-FLUX: COMPATIBILITY WITH COLUMN-SPARSE ACCELERATORS

Finally, we examine HiCache on top of Chipmunk-Flux, a column-sparse accelerator for FLUX, to verify that HiCache is complementary to sparse/efficient attention rather than a competing alternative. We compare three configurations on flux-dev (1024×1024, 50 steps, 200 prompts): pure Chipmunk-Flux, Chipmunk-Flux+Taylor, and Chipmunk-Flux+HiCache with a single $\sigma$ setting. All accelerated variants share the same interval = 5 and order = 2.

Table 9 reports CLIP, ImageReward, PSNR, SSIM, LPIPS, and approximate per-image latency. Moving from pure Chipmunk-Flux to Chipmunk+Taylor significantly reduces latency but degrades quality, especially in ImageReward and LPIPS. Chipmunk+HiCache with $\sigma = 0.5$ largely recovers the quality of the pure Chipmunk baseline—often slightly exceeding it in ImageReward—while retaining latency close to Chipmunk+Taylor. This suggests that HiCache can serve as a plug-in quality-restoring module on top of column-sparse accelerators.

