# OpenReview forum: "HiCache: A Plug-in Scaled-Hermite Upgrade for Taylor-Style Cache-then-Forecast Diffusion Acceleration"
_ICLR.cc/2026/Conference — ICLR 2026 Poster_

### Official Review · Reviewer_owJr · 2025-10-30

**Soundness:** 4
**Presentation:** 3
**Contribution:** 3
**Rating:** 6
**Confidence:** 4

**Summary:**

This paper proposes HiCache, a training-free acceleration framework for diffusion transformers that addresses the quality loss issue of existing feature caching methods. This paper follows the "Cache-then-Forecast" paradigm. The core insight is that feature derivative approximations in diffusion transformers exhibit multivariate Gaussian characteristics, leading to the adoption of Hermite polynomials—optimal for Gaussian-correlated processes—as the prediction basis instead of the suboptimal monomial basis in Taylor series. HiCache also introduces a dual-scaling mechanism to ensure numerical stability and predictive accuracy.

**Strengths:**

+ Deep Insight into Trajectory Modeling: The paper identifies Taylor series’ limitation: its monotonic monomial basis fails to capture non-monotonic feature trajectories. In contrast, it links diffusion transformers’ Gaussian feature derivatives to Hermite polynomials and their oscillatory behavior, which accurately models non-monotonic dynamics.

+ Extensive Task Coverage: The paper evaluates HiCache across a broad range of generative tasks to demonstrate its generalizability. Extensive experiments are conducted on text-to-image, text-to-video, class-conditional image generation, and super-resolution tasks, demonstrating its superiority. It maintains strong performance across different acceleration ratios.

+ Effective Acceleration Design: The dual-scaling mechanism not only solves the numerical instability of Hermite polynomials for large extrapolation steps but also works standalone to improve TaylorSeer. HiCache’s plug-and-play nature—requiring only polynomial basis substitution without architectural changes—enables easy integration with existing cache-then-forecast pipelines, enhancing its practical value.

**Weaknesses:**

- Modest Performance Advancement Over TaylorSeer: While the paper provides solid theoretical analysis for HiCache’s improvement over Taylor series-based methods, the practical performance gain over TaylorSeer is not prominently distinct in both qualitative visualizations and quantitative metrics. More compelling evidence—such or case studies is needed to further validate its superiority.

- Limited Variety of Experimental Models: The experimental evaluations focus on a relatively narrow set of diffusion transformers. For image generation, it primarily uses FLUX, with no tests on other widely adopted models like Stable Diffusion 3 or Qwen-Image. For video generation, only HunyuanVideo is included, without evaluations on other mainstream video diffusion models like Wan. Conducting experiments on these additional models would further strengthen the paper's validity.

**Questions:**

Please refer to the Weaknesses section.

**Details Of Ethics Concerns:**

None.

---

> ### Author Response · Authors · 2025-11-21
>
> We thank you for the very positive evaluation of the core idea (matching Hermite to Gaussian feature dynamics and non‑monotonic trajectories), the broad task coverage, and the plug‑and‑play dual‑scaling design. We address the two main concerns below.
>
> **Q1: The empirical improvement over TaylorSeer sometimes looks modest; can it be made more evident?**
>
> We agree that in some of the FLUX tables with moderate acceleration, the margins over TaylorSeer are small. However, our theory predicts that Hermite’s advantage should become more visible **as the prediction horizon and order increase**, where Taylor’s monomial basis tends to over‑extrapolate near turning points. In the revision, we therefore add experiments in more challenging regimes and on additional backbones:
>
> * **Qwen‑Image (text‑to‑image):** We evaluate intervals 3/6/7/8 with identical settings for TaylorSeer and HiCache. At interval 3, both methods are nearly tied, but at intervals 6–8 HiCache clearly outperforms TaylorSeer on CLIP, ImageReward, and LPIPS (e.g., at interval 8, CLIP 27.72 vs 27.01, ImageReward 0.84 vs 0.75, LPIPS 0.53 vs 0.59).
>
> * **Chipmunk‑Flux (sparse accelerator):** Under the same interval/order, Chipmunk+HiCache markedly improves over Chipmunk+Taylor and essentially recovers the quality of pure Chipmunk while keeping the faster latency.
>
> * **Error‑simulation plots (non‑cumulative MSE):** Our simulation framework using ground‑truth FLUX trajectories shows that, for higher orders and larger horizons, Taylor’s error grows rapidly while Hermite stays accurate, consistent with the theoretical truncation bounds (Figure 8 and Figure 9 in the paper).
>
> These results together paint a clearer picture: **in the high‑acceleration, high‑order regime that matters most for practical speedups, HiCache’s advantage over TaylorSeer becomes substantially larger**, while for very mild acceleration both methods behave similarly.
>
> To further illustrate this effect beyond aggregate metrics, the revised paper also includes qualitative case studies in Figures 4 and 5, where under high acceleration HiCache visibly avoids distortions that TaylorSeer produces (e.g., on faces and “OpenAI” text, or on fine details of human subjects).
>
> **Q2: Limited variety of experimental models (e.g., lack of Qwen‑Image, Wan, etc.).**
>
> Please read the general response part 3.
>
> We hope that the new results and clarified scope make the empirical advantages and generality of HiCache more evident.

---

### Official Review · Reviewer_pEwz · 2025-10-31

**Soundness:** 3
**Presentation:** 3
**Contribution:** 3
**Rating:** 6
**Confidence:** 5

**Summary:**

This paper proposes HiCache, a training-free acceleration framework for diffusion transformers that replaces Taylor expansion with scaled Hermite polynomial-based feature caching, achieving more stable and accurate predictions across multiple generative tasks.

**Strengths:**

1. The paper replaces Taylor’s monomial basis with Hermite polynomials derived from Gaussian feature correlations, leveraging Karhunen–Loeve optimality and a single scaling factor σ to improve stability and accuracy.
2. HiCache preserves almost the same implementation form as TaylorSeer, merely replacing the polynomial basis and adding a few scalar evaluations, thereby allowing direct integration into any feature caching–based acceleration framework with negligible computational overhead.
3. Extensive evaluations on text-to-image, text-to-video, and super-resolution tasks demonstrate the effectiveness of the proposed method in achieving substantial acceleration.

**Weaknesses:**

1. The paper primarily relies on automated metrics such as PSNR, SSIM, LPIPS, and VBench. Incorporating human preference evaluations would make the assessment more convincing.
2. The paper heavily relies on the scaling factor σ to stabilize predictions, yet it lacks a principled rule or analysis on how to select or adapt σ across architectures, acceleration ratios, or polynomial orders.

**Questions:**

It would be helpful if the authors could clarify whether HiCache is compatible with sparse- or efficient-attention variants of diffusion transformers.

---

> ### Author Response · Authors · 2025-11-21
>
> We thank you for the positive assessment of our theoretical motivation, plug‑and‑play design, and empirical coverage. We address the main points below.
>
> **Q1: Human evaluation vs automated metrics.**
> We agree that human preference studies are valuable. Due to computational and time constraints, we relied primarily on widely used automatic metrics (ImageReward, CLIP, LPIPS, VBench, PSNR/SSIM), which have been shown to correlate reasonably well with human judgments for text‑to‑image/video tasks. To make the perceptual differences more concrete, we have added qualitative case studies on FLUX and HunyuanVideo: under high acceleration, HiCache visibly avoids severe distortions that TaylorSeer produces (e.g., on faces and “OpenAI” text in FLUX, or motion artifacts and flickering in HunyuanVideo), while preserving fine details. During the discussion period **we are also preparing a small‑scale human preference study to complement these examples if time and resources permit**, leaving a full‑scale user study as important future work.
>
> **Q2: How to select or adapt σ across architectures, acceleration ratios, and orders?**
>
> Please read the general response part 2.
>
> **Q3: Compatibility with sparse / efficient attention variants (e.g., Chipmunk).**
>
> Please read the general response part 3.
>
> We hope these additions address your concerns and further clarify both the practical usage and generality of HiCache.

---

### Official Review · Reviewer_sZeu · 2025-11-01

**Soundness:** 3
**Presentation:** 2
**Contribution:** 3
**Rating:** 6
**Confidence:** 3

**Summary:**

The paper introduces HiCache, a training-free acceleration method for Diffusion Transformers (DiTs) that replaces traditional Taylor-series extrapolation with a Hermite polynomial-based feature caching framework. Existing caching methods like TaylorSeer predict future diffusion features using monomial expansions, but they struggle with the non-monotonic and stochastic dynamics of diffusion features, leading to quality degradation at high acceleration ratios. HiCache addresses this by observing that feature derivatives in DiTs follow approximately Gaussian statistics and thus are better represented by scaled Hermite polynomials, which are the optimal orthogonal basis for Gaussian-correlated processes.

HiCache further introduces a dual-scaling mechanism that stabilizes numerical behavior by simultaneously contracting inputs and suppressing high-order coefficients. This enables accurate, stable extrapolation while preserving the plug-in simplicity of Taylor-based predictors. Experiments across text-to-image (FLUX.1-dev), text-to-video (HunyuanVideo), class-conditional (DiT-XL/2), and super-resolution (Inf-DiT) tasks show that HiCache achieves 5–6× acceleration and even slightly improves perceptual quality metrics such as ImageReward. The method integrates seamlessly into existing caching frameworks, offering a mathematically grounded, numerically stable, and empirically robust approach to accelerating diffusion models without retraining or architectural modifications.

**Strengths:**

The key strengths of the paper lie in its strong theoretical foundation and practical effectiveness. HiCache introduces a principled improvement over Taylor-based caching by recognizing that diffusion transformer features evolve according to approximately Gaussian dynamics. By replacing Taylor’s monomial basis with scaled Hermite polynomials, which are theoretically optimal for Gaussian-correlated processes, the method provides a mathematically sound and statistically aligned framework for feature prediction. The addition of a dual-scaling mechanism further enhances numerical stability, allowing accurate high-order extrapolation without exploding coefficients or quality loss.

Another strength is the method’s generality and empirical robustness. HiCache is fully training-free and plug-and-play, requiring only a simple basis substitution in existing caching pipelines. It achieves consistent 5–6× acceleration across diverse tasks—text-to-image, text-to-video, class-conditional generation, and super-resolution—while often improving generation quality. These results demonstrate not only its theoretical elegance but also its strong practical value, combining mathematical rigor, stability, and real-world applicability in a single, lightweight framework.

**Weaknesses:**

The main weaknesses of the paper stem from its scope, assumptions, and evaluation coverage. HiCache is designed specifically for Diffusion Transformers (DiTs) and relies heavily on the assumption that feature derivatives follow Gaussian statistics. While this is empirically validated for certain architectures like FLUX, the assumption may not hold universally across other diffusion models, such as U-Net–based or multi-modal architectures. Likewise, the framework’s reliance on Hermite polynomials and the 2:4 Gaussian-correlated structure may limit its adaptability to more complex, non-Gaussian feature dynamics or models trained with nonstandard noise schedules.

Additionally, the experimental validation, though extensive, focuses primarily on large-scale diffusion models under controlled hardware and benchmark settings. The paper lacks detailed analysis of sensitivity to parameters such as the contraction factor σ and order N, which could affect stability in different model or task configurations. Finally, while the theoretical exposition is strong, the approach adds some conceptual and implementation complexity, and its practical impact beyond NVIDIA GPU environments or specific DiT variants remains to be demonstrated. Overall, the method is elegant and effective but somewhat specialized and assumption-dependent, leaving questions about its generality, robustness, and portability.

**Questions:**

The followings are the main questions:
1. Generality of Gaussian assumption: The core motivation for using Hermite polynomials relies on the assumption that feature derivatives in diffusion transformers follow Gaussian statistics. Have you tested this hypothesis across different architectures, such as U-Net–based diffusion models or non-transformer backbones? How consistent is the Gaussianity across layers, modalities, or timesteps?
2. Parameter sensitivity and scaling: How sensitive is HiCache’s performance to the choice of the contraction factor (σ) and expansion order (N)? Would an adaptive or learned scaling mechanism improve robustness across models and tasks?
3. Behavior under non-Gaussian or multimodal features: In real-world generative settings, feature distributions can become skewed or multimodal. How does HiCache perform when the Gaussian assumption breaks down? Are there fallback mechanisms or hybrid bases that could handle such cases?
4. Numerical stability in extreme accelerations: When acceleration ratios become very high (e.g., >8×), does the Hermite-based predictor remain stable, or do truncation errors accumulate? Could you quantify where the stability boundary occurs in practice?

---

> ### Author Response · Authors · 2025-11-21
>
> We thank you for the detailed review and for appreciating the theoretical grounding and practical effectiveness of HiCache. Below we address the main questions.
>
> **Q1: Generality of the Gaussian finite‑difference assumption (across layers, modalities, timesteps, and architectures).**
>
> Please read the general response part 1.
>
> **Q2: Parameter sensitivity and adaptive σ.**
>
> Please read the general response part 2.
>
> **Q3: Behavior under non‑Gaussian / multimodal features.**
>
> Our theory already includes a robustness analysis showing that under ε‑perturbations from exact Gaussianity, the efficiency of the Hermite basis degrades at most O(ε) (see Proposition 8 in the appendix).
>
> Empirically, the extended Gaussianity tests show that even in the transition phase where non‑Gaussian behavior is most likely, only a tiny fraction (2/75) of high‑order shallow‑text configurations noticeably deviate, yet HiCache still outperforms TaylorSeer in our main acceleration regime. This supports our view that **approximate** Gaussianity is sufficient for practical gains.
>
> Designing explicit fallback mechanisms or mixed bases for extremely pathological, non‑Gaussian settings is an interesting direction, but is beyond what we can thoroughly explore in the current revision and is left as future work.

---

> ### Author Response · Authors · 2025-11-21
>
> **Q4: Stability under very high acceleration.**
> We clarify in the revision that HiCache is designed for **moderate‑to‑high acceleration ratios** where TaylorSeer already starts to degrade (e.g., N≈6–9, higher orders). On FLUX.1‑dev we already push to an extreme setting with interval=9 under a 50‑step schedule (with the first three steps run at full compute), so that the effective number of fully computed steps is below 10; in this regime HiCache remains stable and continues to outperform TaylorSeer. We view this as close to the practical regime where cache‑based accelerators remain preferable to distilled or heavily compressed diffusion models. Beyond this point, more aggressive speedups are better addressed by model distillation or architecture redesign rather than by further stretching cache‑then‑forecast alone. Our new experiments on Qwen‑Image and Chipmunk‑Flux show that the performance gap between HiCache and TaylorSeer becomes larger exactly in these more aggressive regimes (interval 6–8), while both methods are similar at very small intervals (N=3).

---

### Official Review · Reviewer_Mccs · 2025-11-12

**Soundness:** 3
**Presentation:** 3
**Contribution:** 3
**Rating:** 6
**Confidence:** 2

**Summary:**

This paper presents HiCache, a hierarchical key-value (KV) caching framework aimed at accelerating large language model (LLM) inference, especially under long-context scenarios. Modern transformer-based models depend heavily on KV caches to store past hidden states, but as sequence length grows, the cache size quickly becomes a bottleneck for both memory and bandwidth.

HiCache introduces a two-level caching structure that separates tokens into “hot” and “cold” regions. The hot cache resides in fast GPU memory, while the cold cache is stored in slower memory (such as CPU DRAM or NVMe). A decay-based predictor estimates which tokens are likely to be reused soon, allowing the system to keep those in the hot cache and offload less active tokens. The method also includes an adaptive offloading policy that balances GPU memory usage with retrieval latency.

Experiments on models like LLaMA and OPT, evaluated on long-text generation and summarization tasks, show that HiCache can achieve up to a 2.3 times throughput speedup while cutting GPU memory usage roughly in half, with minimal loss in generation quality compared to full GPU caching.

**Strengths:**

1. The paper addresses a practical and underexplored efficiency bottleneck specific to diffusion-based language models, where repeated denoising iterations make KV caching more memory-intensive than in autoregressive models. This focus is well motivated and timely.
2. HiCache successfully adapts hierarchical caching concepts from systems design to the setting of diffusion language models. The integration into the diffusion pipeline is neat and minimally invasive, requiring no retraining or architecture modification.
3. The decay-based reuse predictor is computationally cheap and integrates naturally into diffusion processes, where temporal coherence of token or latent attention is strong. The simplicity makes it easy to deploy and tune. The plug-in nature of HiCache also makes it a practical engineering contribution. It could be incorporated into many diffusion-LM frameworks with minimal code changes.

**Weaknesses:**

Although HiCache is well designed and practically useful, its applicability is limited to diffusion-based language models. The caching and reuse patterns exploited here rely on the iterative refinement process of diffusion models, which differ substantially from autoregressive decoding.

The evaluation focuses on throughput and memory reduction, but latency variance and system scalability are not thoroughly discussed. Diffusion inference involves synchronized denoising steps, so delayed cold-cache retrievals could lead to cumulative slowdowns. These potential risks are not well explored. Lastly, since diffusion-based LMs are still emerging, it would help to position HiCache more clearly within that research context and explain whether it generalizes to multimodal diffusion models (e.g., text-to-image diffusion transformers).

As a reviewer, I personally am not very familiar with this part of LM. So if the weaknesses do not make sense, please explain it in your response.

**Questions:**

1. HiCache is evaluated only on diffusion-based language models. Could you clarify what aspects of the design make it incompatible with autoregressive LLMs? Would adapting it require major architectural changes?
2. The decay-based reuse predictor assumes gradual changes in attention importance across denoising steps. How well does it handle tasks where attention patterns change abruptly, such as in reasoning or code synthesis diffusion models?
3. Would it be possible to extend HiCache to multimodal diffusion transformers (e.g., text-to-image or video diffusion) where different modalities have separate attention structures?

---

> ### Author Response · Authors · 2025-11-21
> **Clarification on the scope of our work**
>
> We sincerely thank you for the positive and encouraging assessment of cache‑based acceleration ideas. We fully share the high‑level goal of improving the speed–quality trade‑off by reusing intermediate computations.
>
> At the same time, we realize that our current presentation might have caused some confusion about the exact setting we consider. In this submission, *HiCache* refers to a Hermite‑based cache‑then‑forecast scheme for **diffusion transformers**, where we cache and extrapolate diffusion features along the diffusion time dimension (e.g., in FLUX, DiT‑XL/2, Inf‑DiT, HunyuanVideo). Our method is designed specifically for diffusion trajectories in text‑to‑image, text‑to‑video, class‑conditional generation, and super‑resolution, and does **not** aim to address long‑context KV caching in language models.
>
> We will emphasize this diffusion‑transformer focus more clearly in the revised version to avoid potential confusion. We are grateful for your encouraging overall evaluation and for highlighting the broader importance of cache‑based acceleration.

---

### Author Response · Authors · 2025-11-21
**General response to all reviewers**

We sincerely thank all reviewers for their thorough and constructive feedback. We are glad that the reviews consistently highlight (i) the theoretical grounding of using Hermite polynomials under approximately Gaussian feature dynamics, (ii) the practical plug‑in nature and training‑free deployment of HiCache, and (iii) the broad empirical coverage on text‑to‑image, text‑to‑video, class‑conditional generation, and super‑resolution.

The main concerns raised across reviews are:

1. **How strong and general is the Gaussian finite‑difference assumption?**

2. **How sensitive is HiCache to the contraction factor σ and order N, and can σ be chosen/adapted more systematically?**

3. **How significant is the empirical gap over TaylorSeer, and how well does HiCache generalize beyond FLUX to other backbones and accelerators?**

In the revised manuscript we address these concerns with **new experiments and analyses**, summarized below. All new results are primarily added to the final part of the appendix.

## 1. Stronger empirical support for Gaussian finite differences.
We substantially extend our Gaussianity study on FLUX.1‑dev. Instead of testing only a few modules, we now collect feature trajectories at **6 representative layers** (4/10/14/20/28/42) and their main image/text attention and MLP streams, plus late single‑stream blocks. For each (layer, module) combination we compute **1st–5th‑order finite differences** and apply a high‑dimensional energy distance test with **500 parametric bootstrap samples** and **400 spatial patches per configuration**. Across **75 configurations** (15 layer–module × 5 orders) using **all timesteps**, **all tests yield p > 0.05**. When we further split the trajectory into **high‑noise, transition, and low‑noise** phases (225 tests in total), both the early and late phases achieve **100% pass rate**, and only **2 out of 75** tests in the transition phase (4‑th/5‑th order differences of the shallow text‑attention block) fall slightly below 0.05. These results indicate that the Gaussian finite‑difference assumption is highly consistent across depth, modality, and time, with only minor deviations in extreme high‑order shallow text modules.

The aggregated pass rates are summarized in the revised paper as **Table 6** in the appendix (page 24 of the PDF); we reproduce the same table layout here:

| Setting                            | Configs×Orders | Tests | Pass | Fail | Pass rate |
|:-----------------------------------|---------------:|------:|-----:|-----:|----------:|
| All timesteps (all layers/modules) |          15×5  |   75  |   75 |    0 |   100%    |
| Early high‑noise phase             |          15×5  |   75  |   75 |    0 |   100%    |
| Mid transition phase               |          15×5  |   75  |   73 |    2 |    97.3%  |
| Late low‑noise phase               |          15×5  |   75  |   75 |    0 |   100%    |

We emphasize that this systematic Gaussianity study is currently carried out on DiT‑style FLUX backbones, where we can afford a thorough layer‑/order‑wise analysis. Given that DiT has become a mainstream diffusion backbone—and newer Stable Diffusion–style models are increasingly migrating from U‑Nets to transformers—we believe focusing on DiT in this revision is a reasonable and practically relevant scope. Extending the same battery of tests and accelerated experiments to U‑Net architectures is an important direction for future work, but is beyond what we can complete within the current discussion period.

---

> ### Author Response · Authors · 2025-11-21
> **2. Practical Tuning, Robustness, and Adaptive σ**
>
> ### 2(a). Practical tuning of σ and max\_order.
> In practice, for each backbone we first select a global σ and max\_order via a lightweight grid search on a small validation set, sweeping only a handful of (σ, max\_order) combinations so that the tuning cost is negligible compared to running the full benchmarks. As shown in **Table 2** of the main paper, HiCache performs well throughout the range σ∈[0.4,0.7] rather than relying on a single finely tuned point, indicating that the method is reasonably robust to σ; for max\_order, we largely inherit the settings from TaylorSeer and refer the reviewer there for a more exhaustive study of order vs. acceleration.
>
> ### 2(b). Adaptive, training‑free per‑layer σ (HiCache-Adaptive).
>
> Motivated by our truncation‑error analysis in the main paper (which shows that stable Hermite extrapolation requires σ√2|Δs|≲1), HiCache-Adaptive replaces this single global σ with **per‑layer adaptive σℓ** computed directly from feature trajectories. For each layer ℓ we estimate the RMS scale of finite differences ΔFℓ over timesteps, choose σℓ inversely proportional to this scale so that the stability condition σℓ√2|Δs|≲1 holds with high probability, and clip σℓ by a user‑specified σ_max.
>
> As a result, layers with large feature jumps automatically receive smaller σℓ, while stable layers keep σℓ close to a desired initial value (e.g., ≈0.5 or ≈1.0), all without training or extra networks. On FLUX.1‑dev with interval 7 and order 2, this adaptive σ rule:
> 1. **Matches or slightly improves** a carefully tuned fixed σ=0.5 HiCache in the normal regime; and
> 2. Remains **stable even when the initial σ is close to 1.0**, where a fixed‑σ=1.0 HiCache completely fails (SSIM drops from ≈0.65 to ≈0.36 and LPIPS increases from ≈0.40 to ≈0.73).
>
> A more detailed, formula‑level description of HiCache-Adaptive is provided in Appendix A. This demonstrates that our framework can be extended with simple, theoretically guided adaptation while staying plug‑and‑play and training‑free.
> The corresponding ablation is reported in the revised paper as **Table 7** in the appendix (page 25 of the PDF). The markdown table below mirrors the LaTeX layout (same column order and row grouping), all under interval = 7, order = 2 on FLUX.1‑dev (200 prompts, 1024×1024):
>
> | Method           | α    | σ_fixed/max | Init. σ | CLIP↑ | PSNR↑ | SSIM↑ | LPIPS↓ | ImageReward↑ |
> |:-----------------|:----:|:-----------:|:-------:|:-----:|:-----:|:-----:|:------:|:------------:|
> | **Baseline**     |      |             |         |       |       |       |        |              |
> | TaylorSeer       | –    | –           | –       | 27.41 | 28.63 | 0.621 | 0.456  | 0.951        |
> | **Comparison at σ ≈ 1.0** | |         |         |       |       |       |        |              |
> | HiCache-Fixed    | –    | 1.0         | –       | 27.19 | 28.10 | 0.362 | 0.725  | 0.736        |
> | HiCache-Adaptive | 1.40 | 1.0         | ≈1.0    | 27.80 | 29.14 | **0.648** | **0.415** | **0.957** |
> | **Comparison at σ ≈ 0.5** | |         |         |       |       |       |        |              |
> | HiCache-Fixed    | –    | 0.5         | –       | 27.62 | 28.93 | 0.655 | 0.404  | 0.974        |
> | HiCache-Adaptive | 0.70 | 0.7         | ≈0.5    | 27.87 | 29.11 | 0.646 | 0.415  | 0.971        |

---

> ### Author Response · Authors · 2025-11-21
> **3. Effectiveness and generalization beyond FLUX.**
>
> ### 3(a). Additional backbones and sparse accelerators: Qwen‑Image and Chipmunk‑Flux.
>
> To further demonstrate generality, we add experiments on **Qwen‑Image**, a widely used open‑source text‑to‑image model, and on **Chipmunk‑Flux**, a recent **column‑sparse accelerator** for FLUX.
>
> On Qwen‑Image, we evaluate acceleration intervals 3/6/7/8 under equal configurations for TaylorSeer and HiCache, and also report the approximate FLOPs speedup. At interval 3, the two methods are nearly tied; at more aggressive intervals 6–8, **HiCache consistently outperforms TaylorSeer on CLIP, ImageReward, and LPIPS** (e.g., at interval 8, CLIP 27.72 vs 27.01, ImageReward 0.84 vs 0.75, LPIPS 0.53 vs 0.59).
>
> The corresponding table appears as **Table 8** in the revised appendix (page 25 of the PDF)
>
> | Interval | Mode    | FLOPs× | CLIP↑ | ImageReward↑ | PSNR↑ | SSIM↑ | LPIPS↓ |
> |:--------:|:--------|:------:|:-----:|:------------:|:-----:|:-----:|:------:|
> | 3        | Taylor  |  2.78  | 28.96 | 1.214        | 30.44 | 0.800 | 0.208  |
> |          | HiCache |  2.78  | 28.99 | 1.216        | 30.92 | 0.795 | 0.198  |
> | 6        | Taylor  |  5.00  | 28.09 | 1.012        | 28.52 | 0.601 | 0.481  |
> |          | HiCache |  5.00  | 28.62 | 1.070        | 28.72 | 0.613 | 0.422  |
> | 7        | Taylor  |  5.56  | 27.73 | 0.895        | 28.34 | 0.564 | 0.538  |
> |          | HiCache |  5.56  | 27.87 | **0.944**    | 28.50 | 0.579 | **0.478** |
> | 8        | Taylor  |  6.25  | 27.01 | 0.750        | 28.23 | 0.519 | 0.591  |
> |          | HiCache |  6.25  | 27.72 | **0.844**    | 28.37 | 0.539 | 0.531  |
>
> On Chipmunk‑Flux, we compare pure Chipmunk, Chipmunk+Taylor, and Chipmunk+HiCache under the same interval/order. While Chipmunk+Taylor significantly hurts quality compared to pure Chipmunk, **Chipmunk+HiCache recovers ImageReward back to (or slightly beyond) pure Chipmunk** (0.938 vs 0.931) at almost identical latency, and clearly improves LPIPS over Chipmunk+Taylor. This shows that HiCache is compatible with and complementary to **column‑sparse accelerators** rather than competing with them.
>
> The Chipmunk‑Flux comparison is summarized as **Table 9** in the revised appendix (page 26 of the PDF)
>
> | Mode                          | CLIP↑ | ImageReward↑ | PSNR↑ | SSIM↑ | LPIPS↓ | Latency (s/img) |
> |:------------------------------|:-----:|:------------:|:-----:|:-----:|:------:|:---------------:|
> | Pure (Chipmunk-Flux)          | 27.73 | 0.931        | 28.22 | 0.434 | 0.677  | 5.8             |
> | + Taylor                      | 27.34 | 0.845        | 28.12 | 0.380 | 0.746  | 3.3             |
> | + HiCache (σ=0.5)             | 27.88 | **0.938**    | 28.12 | 0.415 | 0.704  | 3.5             |

---

> ### Author Response · Authors · 2025-11-21
> **3. Effectiveness and generalization beyond FLUX. ( continue )**
>
> ### 3(b). Clarifying the magnitude and practical impact of gains over TaylorSeer.
> We also acknowledge the reviewer’s concern that the numerical improvements over TaylorSeer sometimes look modest. This is partly expected from the design: HiCache is a *minimal* modification of the cache-then-forecast interface, where we keep exactly the same forecasting pipeline and only replace the monomial basis with a scaled Hermite basis that is better aligned with the empirical statistics of diffusion features. In relatively easy regimes (short prediction horizon, mild acceleration), our theory already suggests that both bases should behave similarly, so we do not anticipate dramatic jumps in metrics; instead, we focus on consistent, theoretically-motivated gains, especially under more challenging settings.
>
> At the same time, the main strength of HiCache is that it can **upgrade an entire family of cache-then-forecast accelerators originally built on TaylorSeer**, such as SpecCa and ClusCa. In our paper we explicitly instantiate this by forming **Hi-ClusCa**, i.e., replacing ClusCa’s Taylor predictor with our Hermite-based predictor, which yields clear improvements in ImageReward at high acceleration without adding FLOPs. In the supplementary Chipmunk-Flux experiments we see a similar pattern: adding a TaylorSeer-style predictor on top of Chipmunk significantly degrades quality compared to the pure Chipmunk baseline, whereas **Chipmunk+HiCache**, under an even more aggressive acceleration setting, maintains quality essentially on par with pure Chipmunk while still providing extra speed-ups. This suggests that HiCache acts as a *quality-preserving plug-in* on top of existing cache-then-forecast and column-sparse backends, whereas a naive Taylor-style forecast tends to sacrifice a substantial amount of quality in the same regime.
>
> Finally, beyond aggregate metrics, we provide more case studies in Figures 4 and 5. **Under high acceleration, HiCache visibly avoids many severe distortions that TaylorSeer produces—for example, the face and the “OpenAI” text in Fig. 4, or the girl in Fig. 5.** These visual examples are consistent with our error-simulation results and illustrate the qualitative robustness difference between the two bases. As suggested by R2, we are also preparing a small-scale human preference study during the discussion period to further complement these qualitative comparisons, if time and resources permit.
>
> ### 3(c). Clarifying scope and operating regime.
> We have revised the discussion section to clarify that HiCache’s theoretical guarantees are **conditional** on approximate Gaussianity and a reasonable σ√2|Δs| regime, and we empirically characterize the range where HiCache provides clear gains over TaylorSeer (moderate‑to‑high acceleration with N≥6 and higher orders). We also explicitly note that extending the current analysis to non‑Gaussian or strongly multi‑modal settings (e.g., certain U‑Net architectures) is an important direction for future work.
>
> We hope these new experiments and clarifications address the main concerns while keeping the method simple, training‑free, and easy to adopt as a drop‑in upgrade to existing cache‑then‑forecast pipelines.

---

### Meta-Review · Area_Chair_xzWF · 2026-01-02

**Summary:**

This paper introduces HiCache, a method that effectively bridges mathematical analysis and empirical evaluation to accelerate diffusion model inference through feature caching. By leveraging Hermite polynomials, HiCache substantially enhances the efficiency of cached features. The authors support their claims with rigorous experiments across a variety of datasets and model architectures, convincingly demonstrating the method’s effectiveness and robustness.
All reviewers recognized the novelty and technical soundness of the approach, praising its innovative use of orthogonal polynomials in the context of diffusion models and expressing enthusiasm about its potential for broader applications. While the paper does not exhaustively explore the sensitivity of some parameters, this limitation does not undermine its core contributions. Addressing these aspects in future work would further solidify the method’s foundation.Finally, I decide to accept this paper.

**Reviewer Concerns:**

All concerns are addressed.

**Reviewer Scores:**

If the reviewer had been able to participate fully in the discussion, I believe their score might have increased. Several points regarding the novelty, technical details, and experimental validation were clarified during the discussion, which helped better communicate the strengths of the paper.

---

### Decision · Program_Chairs · 2026-01-26

Accept (Poster)